# Coevolution of parental investment and sexually selected traits drives sex-role divergence

Lutz Fromhage[1] & Michael D. Jennions[2]

Sex-role evolution theory attempts to explain the origin and direction of male–female differences. A fundamental question is why anisogamy, the difference in gamete size that defines the sexes, has repeatedly led to large differences in subsequent parental care. Here we construct models to confirm predictions that individuals benefit less from caring when they face stronger sexual selection and/or lower certainty of parentage. However, we overturn the widely cited claim that a negative feedback between the operational sex ratio and the opportunity cost of care selects for egalitarian sex roles. We further argue that our model does not predict any effect of the adult sex ratio (ASR) that is independent of the source of ASR variation. Finally, to increase realism and unify earlier models, we allow for coevolution between parental investment and investment in sexually selected traits. Our model confirms that small initial differences in parental investment tend to increase due to positive evolutionary feedback, formally supporting long-standing, but unsubstantiated, verbal arguments.

[1] Department of Biological and Environmental Science University of Jyvaskyla, PO Box 35, FI-40014 Jyvaskyla, Finland. [2] Ecology, Evolution & Genetics, Research School of Biology, Australian National University, Canberra, Australian Capital Territory 0200, Australia. Correspondence and requests for materials should be addressed to L.F. (email: lutzfromhage@web.de) or to M.D.J. (email: Michael.Jennions@anu.edu.au).

In most animal species, males and females exhibit clear differences in appearance and behaviour. In general, males invest less than females in their offspring, and instead allocate resources to costly traits that elevate their mating success[1]. This mating investment can involve the evolution of weapons when males compete directly for access to mates, or the evolution of ornaments, elaborate advertisement signals and coercive traits when mates compete indirectly to induce females to mate and then use their sperm.

Sex-role evolution theory attempts to explain the origin and direction of male–female differences. The main areas of attention are sex differences in traits that increase mating rates (for example, searching, courting and fighting)[2]; in discrimination between potential mates[3] and in traits that improve offspring survival (for example, gamete provisioning and parental care; jointly referred to as parental investment)[4–7]. A fundamental question is why anisogamy, the difference in gamete size that defines the sexes, generally correlates with large differences in post-fertilization parental investment in the form of extended parental care. Why does greater investment per gamete (that is, producing eggs, which defines females) tend to be associated with providing more parental care? More generally, how is anisogamy linked to sex-specific selection that produces the full spectrum of sex differences that exist across taxa[8]? The answer to this question is intimately related to the trade-off between parental care and competition for mates[9], and the fact that greater parental care by one sex increases the costs of choosiness for the other sex, by reducing the encounter rate with potential mates[10]. Explaining sex differences in parental care is therefore central to explaining sex roles.

According to a popular view rooted in the seminal works of Bateman[11] and Trivers[4], small initial differences in parental investment became subject to a positive feedback: 'If you specialize in competing, you gain most selective advantage by putting more into competing; and the same for caring. And so the divergence widened over evolutionary time, with natural selection proliferating and amplifying the differences, down the generations'[12]. By definition, anisogamy provides the requisite initial difference in caring because gamete provisioning increases zygote survival and can therefore be regarded as a form of care. This sex difference in early investment leads to the general prediction that, all else being equal, females will invest more into subsequent parental care, and, given a trade-off between caring and competing, that males will invest more into being reproductively competitive. On closer inspection, however, this verbal argument does not really explain why greater mean competitiveness in one sex will select for even greater competitiveness[13]. This relationship is simply asserted, and the underlying assumptions are neither made explicit nor properly justified.

In contrast to the idea that small initial differences between the sexes increase over time due to a positive feedback, Kokko and Jennions[13] recently argued that a powerful and general force acts in the opposite direction to hinder sex-role divergence due to a negative feedback. Their argument ran as follows: if the operational sex ratio (the ratio of males to females available to mate at a given time, OSR[14]) becomes male-biased because males spend less time providing parental care, then it is more difficult for males to find mates. A male-biased OSR makes being in the mating pool less profitable (that is, lowers the mean mating rate of deserting males), which then makes it more beneficial for males to prolong the period of parental care (see the conceptual analysis in Fig. 1 of Kokko and Jennions[13]). According to this view, even though anisogamy creates an initial asymmetry in parental investment, egalitarian sex roles are stable unless additional factors come into play (for example, stronger sexual selection on one sex, mixed parentage of broods, or a biased adult sex ratio (ASR)). This line of argument is based on a formal mathematical model[13] and, as a result, now seems to be broadly accepted (for example, cited in the well-known textbook of Davies et al.[15]). Unfortunately the model and the resultant line of argumentation are, we will argue here, incorrect. We rectify these problems with an updated model, explain the flaws in the previous model, and show that the earlier prediction of convergence towards egalitarian care by males and females in a null model (that is, when the only initial difference between the sexes is anisogamy) does not follow. Our revised model still confirms that predictions about how multiple paternity, sexual selection and the ASR are related to sex differences in parental care remain similar. Crucially, however, we argue that previous explanations for the relationship between the ASR and the proportion of care given by males that invoked a direct causal role for the ASR are misleading. Our revised model ensures that the concepts underpinning our understanding of sex-role evolution is not based on demonstrably flawed arguments. The so-called 'Fisher condition'[16] is a consistency requirement that matings and offspring must be accounted for from the perspective of both sexes: because each diploid offspring in a sexual species has a mother and a father; and each heterosexual mating involves a male and a female. We clarify how the Fisher condition applies to the different life history stages at which the sex ratio is measured, namely birth, maturation, adults and in the mating pool (OSR).

In our model, individuals alternate between two possible states, during which they are either available for mating and in the mating pool (time-in) or unavailable (time-out). During time-in, females are subject to mortality rate $\mu_I$ and (if qualified to mate) mating rate $a$; males are subject to mortality rate $\tilde{\mu}_I$ and mating rate $\tilde{a}$ (throughout, male variables are marked with a sperm-like tilde). Upon mating, individuals enter time-out, and provide parental care for duration $T$ or $\tilde{T}$ with corresponding mortality rates $\mu_O$ and $\tilde{\mu}_O$. If still alive they then return to time-in. To vary the mean relatedness between carers and offspring, we specify the number of females and males involved per breeding event as $n$ and $\tilde{n}$. So monandry occurs when $n = \tilde{n} = 1$, and polyandry when $n = 1; \tilde{n} > 1$. To model non-random variation in mating success due to intra- and/or inter-sexual selection, parameters $k$ and $\tilde{k}$ elevate the expected mating rates of individuals to $k$-fold ($\tilde{k}$-fold) that of average individuals of their sex. To increase generality, we define $r$ as the sex ratio at maturation (henceforth, MSR), that is, among individuals that enter time-in for the first time in their life. This definition makes the model applicable without making any assumptions about whether or not juvenile mortality is equal in each sex (as was required by Kokko and Jennions[13] because they defined $r$ as the sex ratio at conception). For an overview of model variables, see Table 1.

Empiricists are familiar with, and possibly even frustrated by, the apparent ease with which modellers can overturn familiar predictions by altering a few key assumptions. It is therefore important to note that our updated initial null model for the evolution of sex differences in parental care retains the same key assumptions as the original model of Kokko and Jennions[13]. For example, there is a temporal trade-off between caring and competing, both sexes have the same mortality costs when caring (or when competing for mates), both sexes provide the same fitness-enhancing benefits to their offspring when they care (contra McNamara and Wolf[17]), and the effects of biparental care are additive (dependent on the total time parents spend caring) rather than synergistic (contra Barta et al.[18]). Despite a similar setup to the earlier model, in our current null model equal levels of care by both sexes is an unlikely outcome of evolution even though the evolution of caring remains equally likely for both sexes (that is, symmetric). This conclusion arises because our null model predicts that the proportion of care provided by each sex is prone to drift (along a line of equilibria ranging from male-only to female-only care), and that unisexual care is

**Table 1 | Overview of notation.**

| Symbol | Meaning |
|---|---|
| $T$ | Female care duration |
| $n$ | Number of females per breeding attempt |
| $k$ | Strength of sexual selection on females (proportion of maturing females qualified to mate) |
| $\mu_I, \mu_O$ | Female mortality during time-in and time-out |
| $x$ | Female competitiveness |
| $r$ | Maturation sex ratio (MSR) |
| $M$ | Mate search coefficient |
| $\alpha$ | Shape coefficient of brood survival function |
| $\gamma$ | Synergy coefficient |
| $\tau$ | Expected care duration per brood provided by a given female (accounting for the possibility that she may die during care) |
| $a$ | Mating rate of female qualified to mate |
| $p$ | Probability to survive a given time-in period (for female qualified to mate) |
| $s$ | Probability to survive a given time-out period (for caring female) |
| $b$ | Brood size |
| $S$ | Brood survival |
| $W$ | Fitness of average female |
| $v$ | Reproductive value of average female |
| $r_O$ | Operational sex ratio (OSR) |
| $r_A$ | Adult sex ratio (ASR) |

Corresponding male symbols are obtained by marking female symbols with a tilde ($\sim$).

selected for if the sexes differ even slightly in the costs and/or benefits of a given level of caring.

This prediction from our initial null model hinges on the assumption that broods benefit to the same extent from a given increment of care by either sex. To relax this assumption, we also present a synergistic null model in which the benefits of care from both sexes are synergistic rather than additive. Assuming that there are synergistic gains from biparental care is often appropriate. For example, it is likely whenever the sexes provide somewhat different forms of care that complement each other (for example, food versus protection). In contrast to our initial null model, this synergistic model predicts that a specific division of care between the sexes will evolve. This outcome additionally allows us to explore the effect that various factors of interest (for example, sex-specific mortality rates) have on the evolution of the proportion of care provided by males. In so doing, we reject some of the arguments made by Kokko and Jennions[13] that invoke a direct causal effect of the ASR on caring.

Kokko et al.[9] reported that the sex providing less parental investment has greater scope for competitive investment (*sensu*[9]), because, for a given proportional increase in mating rate, this sex can profitably accept a relatively greater cost due to a decline in another fitness component (for example, longevity). This finding is suggestive of self-reinforcing selection for greater competitiveness (hence less care) but the implications are difficult to assess because Kokko et al.[9] did not formally model feedback between competitiveness and parental investment. Here we extend and unify a series of earlier models that separately examined the evolution of either competitiveness or parental investment by treating the other trait as fixed (for example, refs 9,13,19,20). We explicitly investigate how parental investment (specifically, an extended period of parental care) coevolves with investment in sexually selected traits ($x$ or $\tilde{x}$) that elevate mating rates (that is, by searching for mates, or bearing weapons and ornaments) but impose mortality costs. We model this coevolution as a feedback loop in which parental care affects the evolution of competitiveness, competitiveness has mortality costs, and mortality affects the evolution of parental care.

Our model shows that a positive feedback can drive sex-role divergence and select for greater investment into caring by the sex that cares more initially. This finding is in agreement with well-known claims based on verbal arguments[4,21], albeit that these were not formally substantiated nor fully explained. We believe that we add much needed biological realism by presenting a model that includes feedback between investment into caring and into competing. The model provides a better representation of verbal arguments about sex-role evolution.

## Results

**Basic model.** If the sexes initially differ only in their care duration ($\tilde{T} \neq T$), their behaviour evolves to a line of equilibria because there is selection for a constant total amount of care per brood (Fig. 1a). Along this line there is, however, no selection towards any particular combination of care durations, so the proportion of care provided by each sex can change by genetic drift (although this is excluded from our model). The line of equilibria is replaced by unisexual care if the sexes differ even slightly in the costs and benefits of caring. For example, if males face stronger sexual selection ($\tilde{k} > k$; Fig. 1b,d) or lower relatedness to the brood than females ($\tilde{n} > n$; Fig. 1c,d), the outcome shifts towards female-only care, even if males initially care for longer. The outcomes of the model (Fig. 1) were obtained using two alternative methods that are presented in equations (3) and (4) and equations (8)–(15). Our null model is therefore that selection favours neither single-sex care nor equal parental care by both sexes. In contrast, Kokko and Jennions[13] (see their Fig. 3a) predicted equal care by both sexes which they attributed to a negative feedback process driven by how the OSR affects mating rates.

**Synergy model and the ASR.** A model with synergistic benefit of care by both sexes generally predicts parental care by both sexes (Fig. 2a–c), but with less male care than female care if males face stronger sexual selection ($\tilde{k} > k$; Fig. 2b) or if males are less closely related to the brood ($\tilde{n} > n$; Fig. 2c). Parental care by only one sex can still arise though if the sexual asymmetry in $k$ and/or $n$ is sufficiently strong (Fig. 2d). Given synergy there is therefore no longer a line of equilibria (compare Figs 1a and 2a), but rather an optimal level of care for each sex that need not be equal (Fig. 2b,c).

The relationship between the ASR and sex differences in parental care was a key feature of Kokko and Jennions[13]. They predicted increasing proportions of male care when the ASR becomes more male-biased through either of two routes: by increasing the sex ratio at maturation, $r$, or by simultaneously decreasing the costs of competing $\mu_I$ and $\tilde{\mu}_I$ (which disproportionally increases male lifespan if males are initially the less-caring sex). In addition to re-examining these two routes to vary the ASR, we facilitated some important general insights by considering sex-specific differences in mortality when caring or competing (see Discussion section).

Intriguingly, the source of variation driving changes in the ASR determined whether or not the ASR is related to the proportion of care provided by males. If the ASR varies because both male mortality rates ($\tilde{\mu}_I$ and $\tilde{\mu}_O$) change, or because the maturation sex ratio ($r$) changes, then the proportion of care provided by males increases as the ASR becomes more male biased (Fig. 3a). However, if the ASR varies solely due to changes in time-in mortality (either for both sexes, or for males only), this does not affect the proportion of male care (Figs 3b and 4b). This result contradicts the paradoxical finding of Kokko and Jennions[13] (see their Fig. 5), that the proportion of care provided by males decreased when competing became costlier. The source of this discrepancy is a conceptual error in their derivation of selection

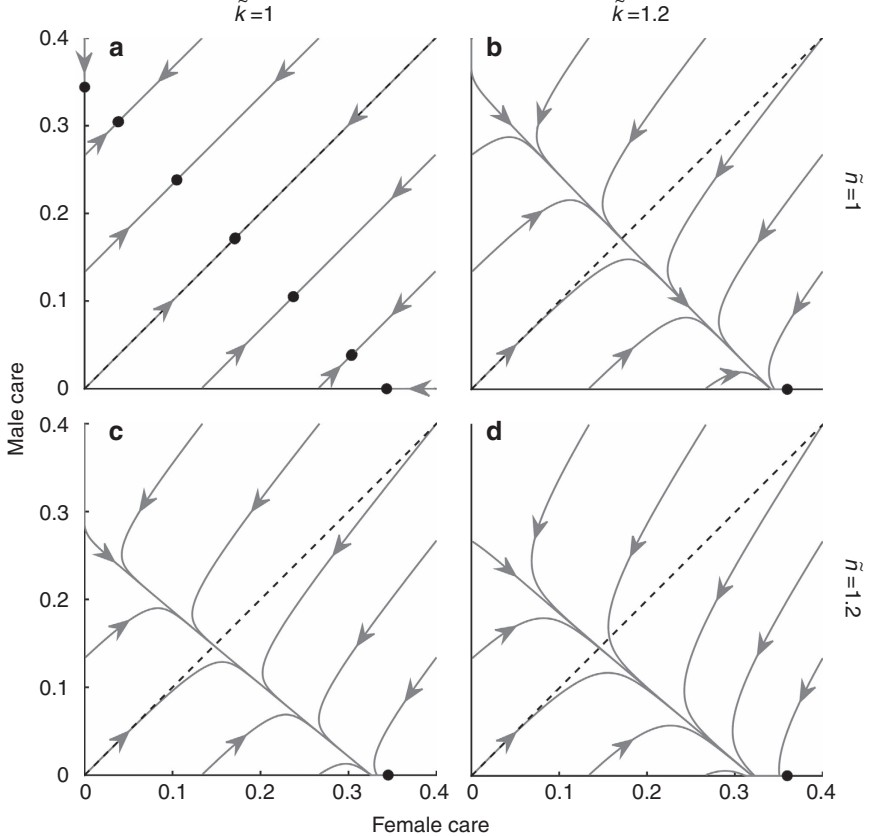

**Figure 1 | Basic model.** Arrows indicate evolutionary trajectories (assuming traits evolve at rates proportional to sex-specific selection gradients) whose stable end points are marked as black dots. Dashed lines indicate equal caring by both sexes. If the sexes differ only in care duration (the null model, (**a**)), trajectories end on a line of equilibria along which there is no selection, at a given total amount of care. If males face stronger sexual selection (**b**), or lower parentage certainty (**c**), or both (**d**), female-only care evolves. Parameter values: $k = n = M = r = 1$; $\mu_O = \mu_I = \tilde{\mu}_O = \tilde{\mu}_I = 0.01$; $\alpha = 0.1$.

gradients (see Methods: Revising Kokko and Jennions' model). In general, the OSR increases with the ASR, but the shape of this relationship depends on the source of ASR variation (Fig. 3c,d).

**Two-trait model.** Synergy can be said to select for egalitarianism in our models because it stabilizes outcomes where both sexes contribute to care. In contrast, coevolution between caring ($T$ or $\tilde{T}$) and competitiveness ($x$ or $\tilde{x}$) has the opposite effect: it can select for uniparental care (Fig. 5) under conditions that would otherwise lead to stable care by both sexes (Fig. 2). If the sexes initially differ only in care duration, the more-caring sex ends up caring alone (Fig. 5a). However, if males face stronger sexual selection ($\tilde{k} > k$) and/or are less related to the brood than females ($\tilde{n} > n$), then female-only care can evolve even if males initially provide more care (Fig. 5b–d). Care by both sexes can always be restored, however, by imposing even stronger synergistic benefits. The non-caring sex exhibits higher competitiveness ($x = 1.59$ in Fig. 5a,d) than the caring sex ($x = 0.9$ in Fig. 5a,c; $x = 0.92$ in Fig. 5b,d) at equilibrium. Although we did not formally model the coevolution of investment into competitive traits ($x$ and $\tilde{x}$) and changes in the strength of sexual selection (that is, $k$ and $\tilde{k}$), it is likely that they will coevolve. It is, however, difficult to know their exact relationship. The same holds for the relationship between sexual selection and the OSR[22]. Exploring these relationships could be a profitable line of future research. Nonetheless, our current model still allows us to determine how changes in sexual selection (arising for whatever reason, including changes in competitiveness) affect patterns

of care. For example, the effect of stronger sexual selection on male care can be seen by comparing the left and right graphs in Figs 1, 2 and 5.

# Discussion
In most taxa, males invest more than females into competing for access to mates (for example, searching, signalling, courting and fighting) and less than females into parental investment (that is, parental care)[2,5,6] (but see refs 1,23 for some notable exceptions). Even if ultimately rooted in anisogamy[8,24], these sex differences are not directly due to anisogamy because gamete size usually accounts for a miniscule part of the total sexual asymmetry in parental investment. The deeper question is what mechanisms link the small initial asymmetry in parental investment, because eggs are larger than sperm, with the subsequent large difference in how much parental care each sex provides. A long-held claim is that small initial differences in parental investment tend to increase over evolutionary time due to a positive feedback process[4,21]. These statements do not by themselves qualify as an explanation for sex roles, however, because they assume the direction of selection *a priori*. Ultimately, they merely re-describe the known pattern, yield no causal insights, and do not explain exceptions to the rule (for example, seahorses and many fish where male-only care has evolved). They replace one unexplained asymmetry with another because they are not explicit about how selection acts[13]. Here we fill this gap with explicit mathematical models to explain the underlying process driving sex differences in care levels.

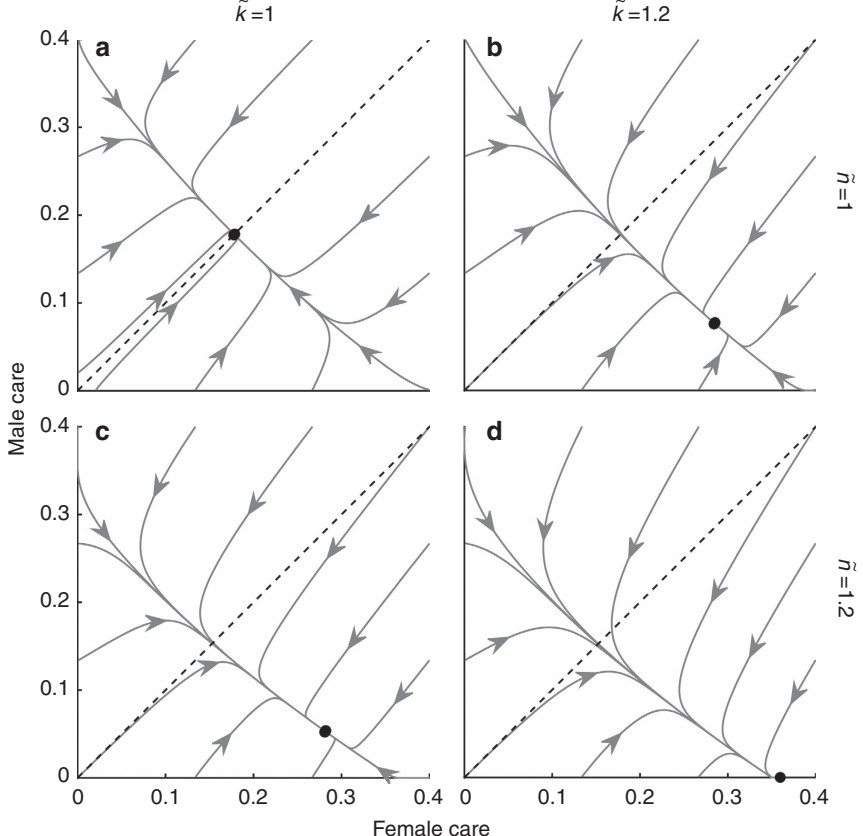

**Figure 2 | Synergy model.** If the sexes differ only in their initial amount of care (**a**), equal care by both sexes evolves. Any deviation from $k = \tilde{k}$ and $n = \tilde{n}$ shifts the stable proportion of care towards the sex which faces stronger sexual selection (**b**), lower parentage certainty (**c**) or both, leading to unisexual care if the asymmetry is large enough (**d**). Parameter values: $k = n = M = r = 1$; $\mu_O = \mu_I = \tilde{\mu}_O = \tilde{\mu}_I = 0.01$; $\alpha = \gamma = 0.1$.

As with many theoretical modelling exercises, a simple null model is the foundation for what follows[25]. Kokko and Jennions[13] built on the insights of Queller[6] to produce a mathematical null model that yielded the key prediction that selection results in a stable equilibrium of equal investment into parental care by both sexes if the only difference between the sexes is anisogamy. Here we overturn this null model. Our most surprising result when re-evaluating Kokko and Jennions[13] is that sex roles do not converge towards equal care, and that equal care is unstable. This instability arises because equal care is either prone to drift (Fig. 1a), or to being replaced by unisexual care if the sexes differ even slightly in the costs and benefits of caring. The reason for this new outcome is that the original analysis contained a conceptual error in identifying the costs of caring (see Methods: Revising Kokko and Jennions' model). We obtained our results using two different methods: one relies on a revised cost–benefit analysis; the other on differentiation of fitness functions (which avoids the difficulty of identifying *a priori* the costs and benefits of caring). Reassuringly, both methods yield identical outcomes.

The practical importance of this new approach is neatly illustrated by looking at how different sources of variation in the ASR affect sex roles. To do so, however, we had to use a second null model that assumes synergistic benefits of care by both sexes. This approach was necessary because this second null model predicts stable levels of care by each sex (Fig. 2), while the additive model does not do so. The synergistic null model confirms earlier predictions[6,13] that males evolve to care less when facing stronger sexual selection (Fig. 2b), reduced share of paternity (Fig. 2c), or both (Fig. 2d).

Kokko and Jennions[13] predicted that a greater proportion of care will be provided by males when the ASR is more male biased. They then argued that the ASR has an independent, causal effect on the net benefit of competing for mates. Unfortunately, it turns out that this claim was misleading: our updated model no longer predicts any effect of the ASR *per se* that is independent of the source of ASR variation. In other words, without specifying the source of ASR variation, we cannot predict the relationship between the ASR and the proportion of male care. We therefore consider it more appropriate to explain causal relationships, and formulate predictions, at the level of the underlying factors affecting the ASR, rather than based on the ASR itself. If the ASR is manipulated by changing the sex ratio at maturation (MSR) then a positive relationship with the proportion of male care does indeed arise (Fig. 3a), confirming the scenario modelled by Kokko and Jennions[13]. This positive relationship occurs because the MSR (which in the absence of sex differences in mortality is equivalent to the ASR) is inversely proportional to male reproductive value. If males have low reproductive value (compared with females, and compared with a given brood), they are more willing to care because they have less to lose in the event of their death. In other words, they are more willing to lay down their own life for the survival of a given brood. The dependence of reproductive values on the MSR follows from the Fisher condition (see Methods: Revising Kokko and Jennions' model). By contrast, a similar relationship between reproductive values and the ASR does not generally hold; adult mortality can affect the ASR without changing the sexes' reproductive values. Intuitively, this occurs because, regardless of adult mortality, an even MSR imposes the constraint that maturing males must on

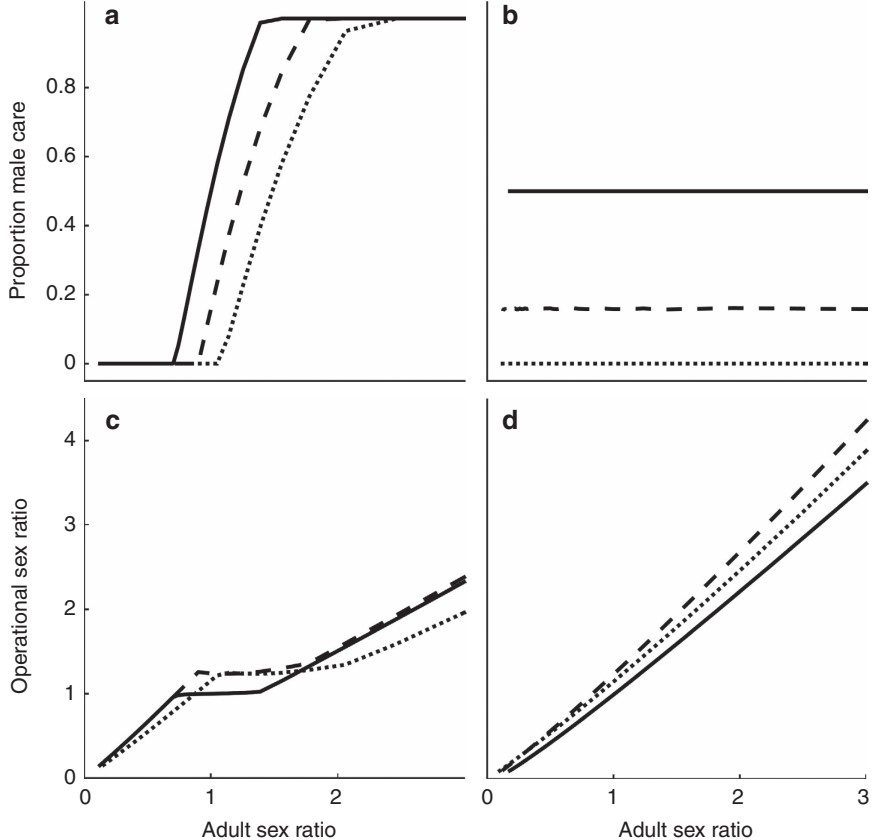

**Figure 3 | Proportion of male care ($\tilde{T}/(\tilde{T}+T)$) and OSR in the synergy model.** (**a**) The proportion of male care and (**c**) the OSR increase with the ASR, when the latter is manipulated either by changing the maturation sex ratio MSR ($r$) or by simultaneously changing male mortality in either state ($\tilde{\mu}_I = \tilde{\mu}_O$). (**b**) The proportion of male care remains constant whereas (**d**) the OSR increases with the ASR, when the latter is manipulated by changing male mortality in time-in ($\tilde{\mu}_I$). Solid line: $\tilde{k}=1, \tilde{n}=1$; dashed: $\tilde{k}=1, \tilde{n}=1.2$; dotted: $\tilde{k}=1.2, \tilde{n}=1.2$. Default parameter values: $k=n=M=r=1$; $\mu_O=\mu_I=\tilde{\mu}_O=\tilde{\mu}_I=0.01$; $\alpha=\gamma=0.1$.

average have the same number of offspring as maturing females. Thus, if (say) males suffer high adult mortality, their lifespan decreases but simultaneously they increase their reproductive rate while alive, generating no overall (dis-)advantage as compared to females. This argument is analogous to the well-known logic of Fisherian sex ratio evolution[26], whereby adult mortality generates no selection on the primary sex ratio as it makes neither sex more valuable than the other.

The same relationship of a more male-biased ASR being associated with a greater proportion of male care also arises if the ASR varies due to a general change in male mortality (both in time-in and time-out) (Fig. 3a). This positive relationship occurs because low mortality makes it cheaper for males to care (equation (13)), by reducing their probability of dying while caring. Simultaneously, because males become more numerous through reduced mortality, male mortality is inversely proportional to the ASR in this scenario.

By contrast, the ASR is uncorrelated with the proportion of male care if it changes solely due to variation in time-in mortality (Figs 3b and 4b). For example, if competing becomes more dangerous for males than females (that is, $\tilde{\mu}_I > \mu_I$), this lowers the ASR but it does not affect the proportion of male care (Fig. 3b). Similarly, if competing becomes equally more dangerous than caring for both sexes (that is, $\tilde{\mu}_I = \mu_I$ increases relative to $\mu_O = \tilde{\mu}_O$) this decreases the ASR whenever males spend more time in the mating pool than females, because they are then more exposed to the elevated risk (Fig. 4c). Again, however, this change in the ASR (despite affecting the OSR; Fig. 4d) does not affect the

proportion of care provided by males (Fig. 4b). This result directly contradicts a similar case study in Kokko and Jennions[13] (see their Fig. 5). It is tempting to predict that males will benefit more from deserting if a female-biased ASR increases mate availability. The incentive to desert sooner is, however, perfectly negated by the greater risk of succumbing to the very mortality that creates the ASR bias. We conclude that time-in mortality does not affect the proportion of male care (Figs 3b and 4b) because it does not affect the reproductive value of males compared to females (see above); nor does it affect the cost of male care (equation (13)) relative to female care (equation (10)). Why then do both sexes care more when it becomes more dangerous to compete (Fig. 4a,b)? This occurs because, as parents expect to raise fewer broods over their lifespan, each brood becomes more valuable. Finally, we note that our model does not consider facultative parental care decisions. The model therefore cannot be tested based on phenotypically plastic shifts in the level of male care in response to, say, mate availability.

Given our findings, how should we interpret recent empirical evidence that a male-biased ASR is associated with a higher proportion of male care in birds[27,28]? Or that parental care is more female-biased in mammals than birds, and that birds more often have a male-biased ASR? There are several possibilities. First, an ASR bias might be caused by a MSR bias. In this case, individuals of the rarer sex have higher reproductive value, and are selected to care less because they have more to lose when risking their life while caring (Fig. 3a). This possibility could be particularly relevant in sexually size dimorphic species, where a negative correlation

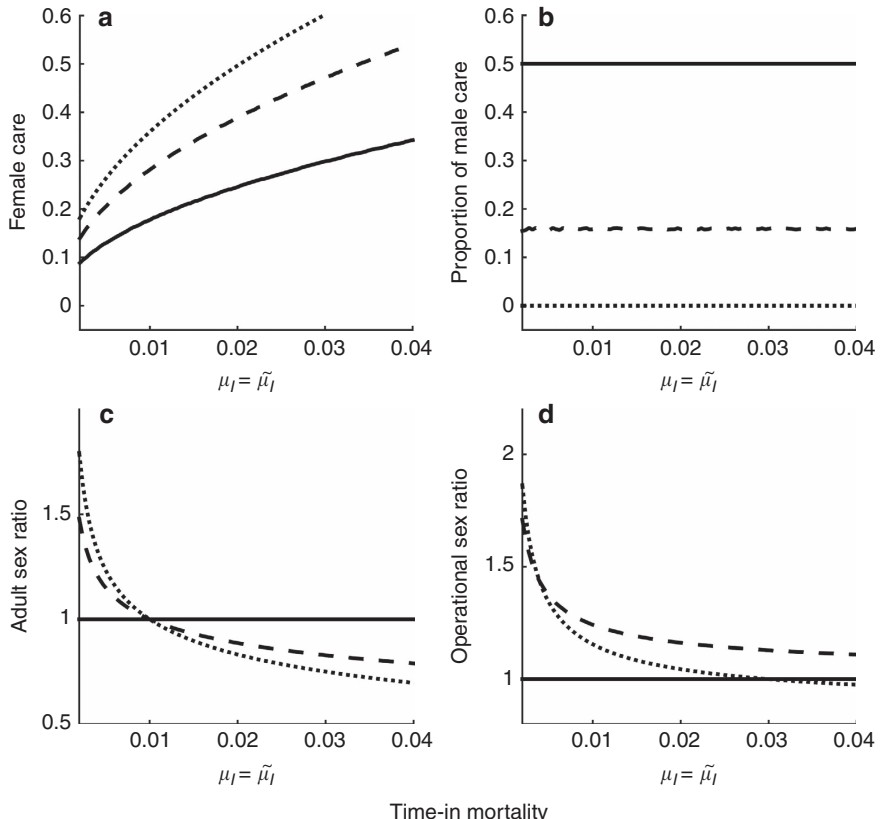

**Figure 4 | Consequences of time-in mortality for both sexes.** (**a**) The duration of female care ($T$) increases with time-in mortality. (**b**) The proportion of male care $\tilde{T}/(\tilde{T} + T)$ remains constant, implying that the duration of male care increases to the same extent as for females. (**c**) ASR and (**d**) OSR decrease with time-in mortality if males are the less-caring sex. Solid line: $\tilde{k} = 1$, $\tilde{n} = 1$; dashed line: $\tilde{k} = 1$, $\tilde{n} = 1.2$; dotted line: $\tilde{k} = 1.2$, $\tilde{n} = 1.2$. Other parameter values: $M = r = 1$; $\mu_O = \tilde{\mu}_O = 0.01$; $\alpha = \gamma = 0.1$.

between body size dimorphism and ASR arises through sex differences in juvenile mortality and/or maturation rates[29,30]. Future studies should focus on estimating the MSR. Second, an ASR bias might arise if one sex generally suffers higher mortality, including while caring, which would reduce its propensity to care (Fig. 3a). Mortality differences could arise if greater competitiveness is associated with sexually selected traits that are costly even outside the context of competing, for example, because they elevate parasite load[31] or raise energetic demands. In general, many morphological sexual traits are likely to elevate mortality even when a male is neither courting nor mate searching (for example, a peacock's train). Third, competitiveness could directly compromise the ability of the more competitive (hence rarer) sex to care, which selects against caring[17]. (Note, however, that when McNamara and Wolf[17] modelled sex differences in the ability to care, they did not consider how competitiveness affects mortality; thus, they predicted the opposite pattern of an ASR bias towards the less-caring sex.)

Our model suggests that any externally driven change in mortality that only arises when competing for mates (for example, an increased density of predators attracted to sexual signals) does not directly select for sex differences in care (Figs 3b and 4b). Causality might, however, work in the opposite direction: if males care less than females because of, say, sex differences in the strength of sexual selection and/or paternity uncertainty (Fig. 2), this selects for greater male competitiveness[9]. This competitiveness, in turn, might manifest in costly sexual traits that elevate mortality and create the observed ASR bias. The future empirical challenge is to disentangle the various possible

explanations for the observed links between the ASR and the proportion of care provided by each sex.

So far we have mostly discussed sex roles as a single trait phenomenon, where care duration is the only trait that can evolve. We have thereby implicitly assumed that traits that elevate an individual's mating rate (that is, are sexually selected) cannot evolve. This assumption is a serious limitation given robust predictions, and abundant empirical evidence, that the sex which provides less parental investment is predisposed to increase its mating rate despite the costs that sexually selected traits impose on other fitness components, such as longevity[9,20]. Indeed, we found that the evolutionary dynamics change profoundly if we allow competitiveness to evolve. It tends to lead to sex-role divergence (Fig. 5) rather than stable levels of care by both sexes (Fig. 2). Intuitively, this result can be explained as follows: an initial difference in parental investment selects for the sex investing less to become more competitive[9,20]. The associated mortality then manifests in care becoming more costly (i.e. a sexual ornament increases mortality while caring, and there is no compensatory benefit at this stage), which selects against care (equations (10) and (13); see also Fig. 3a).

We could also have modelled a scenario where competiveness directly trades-off with the ability to provide care. This scenario would make caring less beneficial for the more competitive sex. The more general point is that an initial difference in parental investment selects for asymmetry in competitiveness, which is then amplified by feeding back on the cost/benefit ratio while caring. Similarly, and perhaps pointing to a general pattern, McNamara and Wolf[17] noted that sex-role divergence occurs

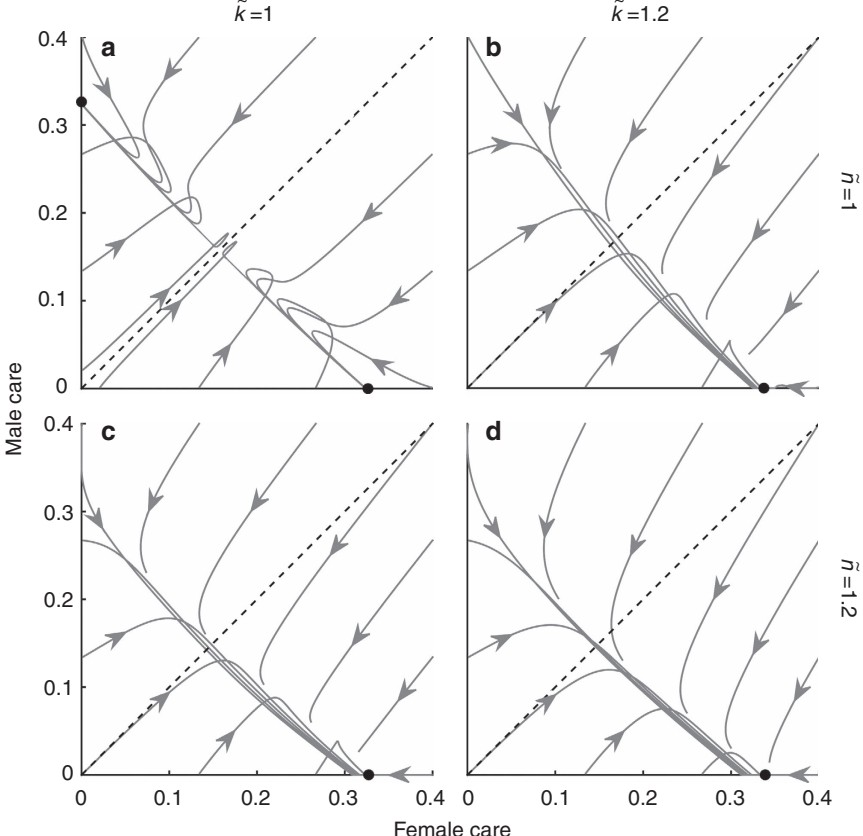

**Figure 5 | Two-trait model in which care coevolves with competitiveness.** If the sexes differ only in their initial amount of care (**a**), the initially more-caring sex ends up caring alone. Sufficiently strong deviations from $k = \tilde{k}$ and $n = \tilde{n}$ shift the outcome towards unisexual care by the sex which faces stronger sexual selection (**b**), lower parentage certainty (**c**) or both (**d**). Parameter values: $k = n = M = r = 1$; $\alpha = \gamma = 0.1$.

more readily when care effort and care ability can both evolve, than when only care effort can evolve.

If anisogamy tends to favour the evolution of female-biased care, how can we explain male-only care in the light of our model? One possibility is that biparental care, driven by synergy, acts as an intermediate stage, followed by environmental changes that affect any of the parameters we have shown to be relevant in this context (the maturation sex ratio, the strength of sexual selection and certainty of parentage). Alternatively, an environmental change could directly reverse the relative time-out durations of the sexes, as documented, for example, in Mormon crickets in response to diet quality[32].

An important criterion for the internal consistency of theoretical models is the so-called Fisher condition. It states that all matings and sexually produced offspring must be accounted for from the perspective of both sexes (after ref. 16). Although this idea sounds simple, there is confusion about how to apply it in specific contexts. Only some sex ratio concepts count individuals that jointly account for all matings and reproduction (fitness) in a population. For example, this is true for the primary sex ratio (at conception), the secondary sex ratio (at birth) and the MSR. In each case, if the ratio is even, then the males and females in question must have equal lifetime expectations of matings and of fitness. Crucially, however, the same is not true for either the OSR or the ASR. In particular, although an even OSR implies equal mating rates (per time unit; see Methods) of males and females in time-in, this need not translate into equal lifetime expectations. To see this, consider a male and female that have just matured, and contribute to both the OSR and ASR: their lifetime expectation of matings and

fitness are equal if the MSR is even, regardless of any subsequent OSR and ASR biases that arise from sex-specific life histories (for example, sex differences in time-out duration or in adult mortality). This distinction highlights the pitfall in Kokko and Jennions'[13] argument that the sex towards which the OSR is currently biased is under selection to provide more care because individuals of the more common sex in the mating pool will reproduce less often. If this statement is interpreted as referring to lifetime expectations of matings and fitness, it does not follow from the Fisher condition. If it is interpreted as referring to mating rates, the conclusion about selection for increased caring does not follow, because neither the cost nor the benefit of caring depends on mating rates (see Methods: Revising Kokko and Jennions' model).

In contrast, in a subsequent review, Jennions and Kokko[10] (p 357) correctly noted (except that MSR should be used, not ASR) that: "If offspring die for lack of care, both parents lose the same number of offspring. Unless the ASR is biased [...], the proportion of an individual's lifetime breeding formed by these offspring is, on average, the same for both sexes because males and females reproduce equally often. It is incorrect to assume that males [or, for that matter, females] can more rapidly compensate for such a loss by remating sooner". The confusion between ASR and MSR in the above quote can plausibly be traced to Queller[6], whose discussion of this topic appears to rest on the implicit assumption that unbiased adult mortality ensures that ASR = MSR. Without this assumption, his discussion, too, needs to be rephrased using MSR rather than ASR.

There is a long history of modellers struggling with the Fisher condition. For example, model 2 of Maynard Smith[33] failed to

specify where the mating opportunities of deserting males came from, creating the impression they appeared from thin air (but see ref. 34, p 128). This problem went unnoticed for many years, until Webb et al.[35] and Wade and Shuster[36] proposed solutions (review: ref. 37), the latter of which turned out to be flawed itself[38]. Kokko and Jennions[13] subsequently drew on Queller[6] and made the Fisher condition the focal point of their model for the evolution of parental care but, as we have described here, they too inadvertently misapplied the Fisher condition. We trust that our current model is now Fisher consistent, although the literature on this topic is hardly a recommendation for uncritical acceptance of such assurances. Our model corroborates the claim that anisogamy can initiate a self-reinforcing selection for asymmetrical parental investment, driven by coevolution with competitiveness. The net strength of selection is then modified, as predicted by Queller[6] and others, by sex differences in the strength of sexual selection, mean relatedness to offspring in a brood/litter, and the degree of synergistic benefits of biparental care.

## Methods

**Basic model.** We follow the assumptions of Kokko and Jennions[13], which can be summarized as follows. Individuals alternate between two possible states, during which they are either available for mating (time-in) or unavailable (time-out). During time-in, females are subject to mortality rate $\mu_I$ and (if qualified to mate; see below) mating rate $a$; males are subject to mortality rate $\tilde{\mu}_I$ and mating rate $\tilde{a}$. Upon mating, individuals enter 'time-out', and provide parental care for duration $T$ or $\tilde{T}$ with corresponding mortality rates $\mu_O$ and $\tilde{\mu}_O$. If still alive they then return to time-in.

To vary the mean relatedness between carers and offspring, we specify the number of females and males involved per breeding event as $n$ and $\tilde{n}$. So monandry occurs when $n = \tilde{n} = 1$, and polyandry when $n = 1; \tilde{n} > 1$. To model non-random variation in mating success due to intra- and/or inter-sexual selection, parameters $k$ and $\tilde{k}$ elevate the expected mating rates of individuals to $k$-fold ($\tilde{k}$-fold) that of average individuals of their sex. One interpretation of our parameters $k$ and $\tilde{k}$ controlling the strength of sexual selection is that some individuals belong to a class comprising $1/k$ of females and $1/\tilde{k}$ of males (at maturation), who are qualified to mate and are therefore the only individuals that will have the opportunity to care. Here we adhere to this strict interpretation to avoid any ambiguity about the meaning of $k$ and $\tilde{k}$. A previously neglected consequence of this interpretation is that the mating rate of each sex is limited by access to opposite-sex individuals that are qualified to mate, rather than all opposite-sex individuals. We therefore define the operational sex ratio $r_O$ as the ratio of qualified males and females in time-in, replacing $r$ with $r \cdot k/\tilde{k}$ in equation A9 of Kokko and Jennions[13] (yielding our equation (18)). This modification ensures consistency for any combination of $k$ and $\tilde{k}$, by linking the mating rates $a = Mnr_O^{1/2}$ and $\tilde{a} = M\tilde{n}r_O^{-1/2}$ of individuals qualified to mate such that $a/\tilde{a} = r_O n/\tilde{n}$, as required by the 'Fisher condition'[16] (see Discussion). Here, the parameter $M$ accounts for species-specific factors such as population density and movement capacity. In equation (23), we also present an alternative method of deriving $r_O$, based on applying the Fisher condition to fitness rather than mating rates. Both methods yield identical results, as should be the case in a fully consistent model.

For mathematical convenience it is desirable to treat all broods of a given parent as equivalent in terms of how much care they receive (because the total amount of care received affects the survival prospects of offspring). Kokko and Jennions[13] achieved this outcome by assuming that broods fail completely unless all carers survive and provide the duration of care prescribed by their strategy. This assumption ensures that the total amount of care received by viable broods is constant. Here we do not follow this assumption because it introduces an unwanted and biologically unrealistic asymmetry: if a focal carer's effort goes to waste whenever another carer dies, this wastage will disproportionately affect the less-caring sex, whose mates are more likely to die during their (longer) care period. Instead, since we are interested in a strategy's average effect per brood, we define average brood survival as a function of the average amount of care per brood. This assumption ensures that offspring benefit equally from additional care received from either parent, which allows us to follow Kokko and Jennions[13] in focussing on other factors. (This is a deliberate simplification; in reality, there may be many reasons—including some related to sex-specific care durations—why offspring might benefit more from additional care by one parent or the other.) Specifically, we assume that offspring survive to enter the time-in state with probability $S[\tau_{total}] = \exp[-\frac{\alpha}{\tau_{total}}]$, which is an increasing function with diminishing returns of the expected total care duration, $\tau_{total} = n\tau + \tilde{n}\tilde{\tau}$. Here, $\tau[T] = T - \int_0^T e^{-t\mu_O}\mu_O(T-t)dt = (1 - e^{-\mu_O T})/\mu_O$ is the expected care duration provided per brood by a female whose strategy is to desert after $T$ (and similarly for males). Here, $e^{-t\mu_O}\mu_O dt$ is the probability that the female dies at time $t$ during her care period, in which case the remaining portion $(T-t)$ of her care period remains unrealized.

For an alternative null model in which expected investment and actual investment per brood are identical (as would occur if investment was pre-committed in, say, the form of gamete size or provisioned food), see section 'A null model for pre-mating parental investment' below.

These assumptions lead to the following fitness functions. A female qualified to mate has probability $p = \frac{a}{a + \mu_I}$ of surviving a given time-in period (once it has started), and probability $s = e^{-T\mu_O}$ of surviving a given care period. Hence $sp$ is the probability that at least one more mating will follow after any given mating; $(sp)^i(1-sp)$ is the probability of mating exactly $i$ additional times; and $p\left(1 + \sum_{i=1}^{\infty} (sp)^i(1-sp)i\right) = \frac{p}{1-sp}$ is the expected number of matings over her lifetime. The lifetime reproductive success (fitness) of an average female is then:

$$W = \frac{1}{k} \cdot \frac{p}{1-sp} \cdot b \cdot S \qquad (1)$$

where $b$ is brood size. Analogously, average male fitness is:

$$\tilde{W} = \frac{1}{\tilde{k}} \cdot \frac{\tilde{p}}{1-\tilde{s}\tilde{p}} \cdot b \cdot \frac{n}{\tilde{n}} \cdot S \qquad (2)$$

The term $n/\tilde{n}$ accounts for shared paternity of the offspring of $n$ females among $\tilde{n}$ males. From this we can calculate selection gradients as proportional derivatives of fitness with respect to a focal trait (p 299 in ref. 39). This method avoids the difficulty of having to identify a priori the fitness costs of care, which is the step where Kokko and Jennions[13] erred (see below). In equation (1) $s$ and $S$ can be expressed as functions of a rare mutant's care duration $T$ (whereas all other variables are treated as constants for the purpose of differentiation), to obtain the following selection gradient for female care:

$$\frac{dW}{dT W} = \frac{S'}{S} + \frac{ps'}{1-ps} \qquad (3)$$

where the derivatives $s' = -e^{-\mu_O T}\mu_O$ and $S' = \frac{dS}{d\tau}\frac{d\tau}{dT}$ (by the chain rule, using $\frac{dS}{d\tau} = \alpha \cdot \exp[-\alpha/\tau_{total}]/\tau_{total}^2$ and $\frac{d\tau}{dT} = e^{-\mu_O T}$), are evaluated at the resident value of $T$. Calculated analogously, the selection gradient for male care is:

$$\frac{d\tilde{W}}{d\tilde{T}\tilde{W}} = \frac{\tilde{S}'}{\tilde{S}} + \frac{\tilde{p}\tilde{s}'}{1-\tilde{p}\tilde{s}} \qquad (4)$$

We then calculate evolutionary trajectories based on the standard assumption that trait values change at rates proportional to these gradients.

To calculate the ASR (denoted $r_A$) in a way that matches the strict interpretation of $k$ and $\tilde{k}$ (but still includes all individuals, whether qualified to mate or not), we revise Kokko and Jennions'[13] equation A10 as:

$$r_A = r\left(\frac{1}{\tilde{k}}(\tilde{L}_I + \tilde{L}_O) + \frac{\tilde{k}-1}{\tilde{k}} \cdot \frac{1}{\tilde{\mu}_I}\right) \Big/ \left(\frac{1}{k}(L_I + L_O) + \frac{k-1}{k} \cdot \frac{1}{\mu_I}\right) \qquad (5)$$

Here, the $1/\tilde{k}$ males qualified to mate spend expected mean times $\tilde{L}_I$ and $\tilde{L}_O$ (see equations (16) and (17)) in time-in and time-out, respectively, while the remaining $\frac{\tilde{k}-1}{\tilde{k}}$ males spend $\int_0^\infty e^{-t\tilde{\mu}_I}t dt/(\int_0^\infty e^{-t\tilde{\mu}_I}dt) = 1/\tilde{\mu}_I$ in time-in.

**Synergy model.** To modify the basic model so that a specific care duration evolves for each sex (see Results section) we next assume that the offspring's probability of reaching maturity depends on a synergistic benefit of biparental (strictly speaking, any form of two sex) care. Offspring survival $S[\tau_{synergy}]$ is now a function of $\tau_{synergy} = \tau_{total}(1 + \gamma n\tau\tilde{n}\tilde{\tau}/\theta)$, where the parameter $\gamma$ specifies the strength of synergy. One unit of egalitarian care corresponds to $(1+\gamma)$ units of uniparental care, and $\theta = n\frac{\tau+\tilde{\tau}}{2}\tilde{n}\frac{\tau+\tilde{\tau}}{2}$ is a normalizing factor. The selection gradients in equations (1) and (2) are now obtained by using

$$\frac{dS}{d\tau} = \frac{\alpha \cdot \exp\left[\frac{-\alpha\theta}{\tau_{total}(\theta + \gamma n\tau\tilde{n}\tilde{\tau})}\right]\theta(\theta + \gamma\tilde{n}\tilde{\tau}(2n\tau + \tilde{n}\tilde{\tau}))}{\tau_{total}^2(\theta + \gamma n\tau\tilde{n}\tilde{\tau})^2}$$

(and analogously for $dS/d\tilde{\tau}$). This reduces to the basic model when $\gamma = 0$.

**Two-trait model.** To allow for coevolution of investment into parental care and sexually selected traits (that is, a higher mating rate) by each sex, we modify the synergy model by expressing female and male mating rates as functions of their competitiveness $x$ and $\tilde{x}$, as $a = x\tilde{x}Mnr_O^{1/2}$ and $\tilde{a} = x\tilde{x}M\tilde{n}r_O^{-1/2}$ (see ref. 20). The OSR ($r_O$) is now obtained by substituting $M$ with $x\tilde{x}M$ in equation (18) (or, equivalently, in equation (23)). We assume that increased competitiveness is costly and increases mortality: $\mu_I[x] = \mu_O[x] = 0.05(1 + x^{1.5})$ (and analogously for males). This formulation means that the costs of being more competitive are expressed regardless of whether or not an individual is in the mating pool. As such, it is best viewed as the cost of bearing a sexual trait rather than, for example, the cost of courting or mate searching. The values 0.05 and 1.5 carry no special significance, but were chosen such that the basic model is recovered when competitiveness is held fixed at $x = \tilde{x} = 1$. Using $x = \tilde{x} = 1$ as initial values, we can then calculate evolutionary trajectories from the same starting points used in the basic model. In equation (1) $p$ and $s$ can be expressed as functions of the competitiveness $x$ of a rare mutant (while treating $r_O$ as a constant for the purpose of differentiation, since it depends only on resident behaviour). So the selection

gradient for female competiveness is:

$$\frac{\mathrm{d}W}{\mathrm{d}xW} = \frac{p' + p^2 s'}{p - p^2 s} \tag{7}$$

where derivatives are evaluated at the resident value of $x$, and

$$s' = -e^{-\mu_O^T T}\mu_{O}'$$

$$p' = \frac{a'}{a + \mu_I} - \frac{a(a' + \mu_I')}{(a + \mu_I)^2}$$

$$a' = \tilde{x}Mnr_O^{1/2}$$

$$\mu_I' = \mu_O' = 0.075x^{0.5}$$

The selection gradient for male competitiveness is calculated analogously, except

$$\tilde{a}' = xM\tilde{n}r_O^{-1/2}$$

**Revising Kokko and Jennions' model.** Kokko and Jennions'[13] analysis suffered from a conceptual error which affected the calculation of selection gradients based on their equations (1) and (2). First, they reasoned that "the future pay-off from desertion is directly proportional to how soon a parent can mate and leave the mating pool (that is, parental mating rate)"[13] (the first term of their equation (1)). This statement implies that the sequence of events: (brief time-in, long time-out, reproduction) promises a higher pay-off from desertion than the sequence: (long time-in, brief time-out, reproduction) of the same total duration. But this implicit assumption appears unjustified, since the fitness consequences are the same in each case. Second, when subtracting their equation (1) from their equation (2), they invoked this presumed pay-off from desertion as an opportunity cost of caring, representing opportunities lost because the carer did not return to the mating pool sooner. To see why this is incorrect, note that they modelled a population that (at equilibrium) is constant over time. Thus, the pay-off from re-entering the mating pool after breeding must also be constant over time: it makes no difference for a focal individual's future (from the time of entering onwards) if it enters now or later, provided that it does eventually do so. This means that caring is costly only insofar as it involves a risk of death (and hence loss of reproductive value) during the care period. To quantify these costs, we need to know the relevant reproductive values (that is, expectations of future reproductive success).

We derive reproductive values as follows. By the Fisher condition, fitness of females and males at maturation is linked as $W = \tilde{W} \cdot r$. Because the model includes no senescence, individuals entering time-in after breeding are indistinguishable from those entering time-in at maturation; hence reproductive values in time-in are also linked by the Fisher condition. Without loss of generality (because only relative reproductive values matter), we arbitrarily define as unity the average reproductive value of maturing offspring of either sex:

$$\frac{1}{r+1}v + \frac{r}{r+1}\tilde{v} = 1 \tag{8}$$

With probability $\frac{1}{r+1}$ a randomly chosen offspring is female, in which case it has reproductive value $v$; with probability $\frac{r}{r+1}$ it is male, in which case it has reproductive value $\tilde{v}$. From equation (8) and the Fisher condition $v = \tilde{v} \cdot r$, we obtain reproductive values $v = (1+r)/2$ and $\tilde{v} = (1+1/r)/2$ for average females and males in time-in, and $vk$ and $\tilde{v}k$ for individuals qualified to mate.

The Fisher condition also implies that the reproductive value of males (relative to females) is inversely proportional to the MSR. This is true for average males and females (namely, $\frac{\tilde{v}}{v} = \frac{1}{r}$) and also for those qualified to mate (namely, $\frac{\tilde{v}k}{vk} = \frac{1}{r}\frac{k}{k}$). Since the right hand side of these equations depends only on $r$, $k$ and $\tilde{k}$, it follows that the reproductive value of males relative to females is independent of adult mortality, and also of any associated variation in the ASR.

We now use these reproductive values to perform a cost-benefit analysis. To quantify the strength of selection acting on a rare female mutant whose care duration $T$ is slightly longer than that of the resident female population, we calculate the marginal net benefit $\omega[T]$ of caring (rather than deserting) at time $T$, and deserting immediately afterwards. The expected consequences of this mutant behaviour must be calculated from the start of a care period, taking into account the possibility of dying before $T$. (To see why this is necessary, consider the extreme case where $T$ is so long that the carer is unlikely to survive until $T$. In this case, selection is weakened by the fact that any mutant with a slightly different $T$ is unlikely to survive long enough for its mutant phenotype to become expressed.) A female that cares at $T$ can expect to gain reproductive value at rate

$$\text{benefit} = s \cdot \frac{\mathrm{d}S}{\mathrm{d}\tau} \cdot b \cdot \frac{1}{2} \tag{9}$$

In words: if the female is still alive at $T$ (as happens with probability $s$), she confers a marginal survival gain $\frac{\mathrm{d}S}{\mathrm{d}\tau}$ (evaluated at the resident values of $T$ and $\tilde{T}$) on her brood of size $b$, to which she is related by ½. Surviving offsprings' reproductive value does not show up in this equation because it was set to unity (see above). Caring at $T$ has costs in terms of the female's own survival, namely

$$\text{cost} = s \cdot \mu_O \cdot kv \tag{10}$$

In words: if she is still alive at $T$ (as happens with probability $s$), she is subject to mortality $\mu_O$, which puts at risk the reproductive value $kv$ she has when deserting. The expected net benefit of caring at $T$ in a given breeding attempt is then

$$\omega = \text{benefit} - \text{cost} \tag{11}$$

For males, the analogous equations are:

$$\widetilde{\text{benefit}} = \tilde{s} \cdot \frac{\mathrm{d}S}{\mathrm{d}\tilde{\tau}} \cdot b \cdot \frac{1}{2} \cdot \frac{n}{\tilde{n}} \tag{12}$$

$$\widetilde{\text{cost}} = \tilde{s} \cdot \tilde{\mu}_O \cdot \tilde{k}\tilde{v} \tag{13}$$

$$\tilde{\omega} = \widetilde{\text{benefit}} - \widetilde{\text{cost}} \tag{14}$$

where $n/\tilde{n}$ accounts for shared paternity of the offspring of $n$ females among $\tilde{n}$ males. Since the expected net benefits $\omega$ and $\tilde{\omega}$ arise per breeding attempt, to compare selection between the sexes we must also take into account that there are $n$ female breeding attempts per $\tilde{n}$ male breeding attempts. Letting $T$ and $\tilde{T}$ evolve at rates proportional to $n\omega$ and $\tilde{n}\tilde{\omega}$ yields the trajectories in Fig. 1.

Equations (9)–(14) reveal that the direction of selection is the same for both sexes if they differ only in care duration (that is, if $T \neq \tilde{T}$; $r = 1$; $\mu_O = \tilde{\mu}_O$; $n = \tilde{n}$; $k = \tilde{k}$) and there is no synergy (hence more care by either sex has the same effect on brood survival, $\frac{\mathrm{d}S}{\mathrm{d}\tau} = \frac{\mathrm{d}S}{\mathrm{d}\tilde{\tau}}$), because $\omega$ and $\tilde{\omega}$ are then equal (up to the positive factor $s/\tilde{s}$).

The evolutionarily stable care duration depends on how parents value broods compared to their own life. This depends on brood size, which in a population at demographic equilibrium must meet the criterion that births match the number of deaths. Specifically, mature females must on average produce one mature daughter during their lifetime, that is, $W \cdot 1/(r+1) = 1$, where $W$ is taken from equation (1), and $1/(r+1)$ is the proportion of daughters in the maturation sex ratio, $r$. Solving this yields brood size at demographic equilibrium:

$$b = \frac{k(1-ps)(r+1)}{pS} \tag{15}$$

More generally, density dependence might act on brood size and/or juvenile survival, so $b$ can also be interpreted as a product of brood size and a density-dependent juvenile survival probability. In contrast, Kokko and Jennions[13] defined brood size a priori, without accounting for demographic stability. This was inconsistent with their implicit assumption of demographic stability when deriving the OSR.

**Revised expressions from Kokko and Jennions.** Here we reproduce some expressions from Kokko and Jennions[13] (henceforth KJ) in slightly revised form. According to KJ's equation (A2), the expected total duration spent in time-in of an average female is

$$L_I = \frac{1}{a(1 - e^{-\mu_O T}) + \mu_I} \tag{16}$$

However, since we have defined $a$ as the mating rate of females qualified to mate (rather than of average females, as in KJ), in our model this equation describes the total duration that a female qualified to mate spends in time-in. The expression $\tilde{L}_I$ for males is analogous. Similarly, based on KJ's equation (A5), in our model

$$L_O = \frac{a(1 + \mu_O T - e^{\mu_O T}(1 - \mu_O T))}{(\mu_I + a(1 - e^{\mu_O T}))\mu_O} \tag{17}$$

is the total duration that a female qualified to mate spends in time-out. The expression $\tilde{L}_O$ for males is analogous. Finally, using KJ's equation (A9) and replacing $r$ with $r \cdot k/\tilde{k}$ to match our definition of the operational sex ratio $r_O$ as the ratio of qualified males and females in time-in, we have

$$r_O = r\frac{k}{\tilde{k}}\frac{\mu_I}{\tilde{\mu}_I} + \frac{M^2 A}{2\tilde{\mu}_I^2}\left(A \pm \sqrt{A^2 + \frac{4\mu_I\tilde{\mu}_I rk}{M^2\tilde{k}}}\right) \tag{18}$$

Where $A = nr\frac{k}{\tilde{k}}(1 - e^{-\mu_O T}) - \tilde{n}(1 - e^{-\tilde{\mu}_O \tilde{T}})$.

**A null model for pre-mating parental investment.** To test the hypothesis that the timing of parental investment with respect to the time of mating does not fundamentally alter the properties of our null model, here we consider the evolution of parental investment that occurs before (rather than after) mating (for example, gamete investment, nest building). Adult life now begins in the time-out state (at maturation sex ratio $r$) and, to reproduce, individuals must survive a full cycle of time-out and subsequent time-in. Thus, a newly matured female has probability $sp$ (as defined for equation (1)) of producing at least one brood, and probability $(sp)^i(1 - sp)$ of producing exactly $i$ broods, amounting to a lifetime expectation of $\sum_{i=1}^{\infty} i(sp)^i(1 - sp) = \frac{sp}{1-sp}$ broods. This (and the analogous argument for males) leads to fitness functions

$$W = \frac{1}{k} \cdot \frac{sp}{1-sp} \cdot b \cdot S \tag{19}$$

$$\tilde{W} = \frac{1}{\tilde{k}} \cdot \frac{\tilde{s}\tilde{p}}{1-\tilde{s}\tilde{p}} \cdot b \cdot \frac{n}{\tilde{n}} \cdot S \tag{20}$$

and selection gradients

$$\frac{\mathrm{d}W}{\mathrm{d}TW} = \frac{S'}{S} + \frac{s'p}{1-sp} \qquad (21)$$

$$\frac{\mathrm{d}\bar{W}}{\mathrm{d}\bar{T}W} = \frac{S'}{S} + \frac{\tilde{s}'\tilde{p}}{1-\tilde{s}\tilde{p}} \qquad (22)$$

Only individuals that survive their first time-out period will ever enter the mating pool in this version of the model. The sex ratio among individuals entering the mating pool for the first time should therefore depend on time-out durations, rather than being an independent parameter. We account for this by re-defining the maturation sex ratio $r$ as referring to individuals that enter time-out (rather than time-in) for the first time. We must also account for this when deriving the operational sex ratio $r_O$. Observing that female fitness $W[r_O]$ can be expressed as an increasing function of $r_O$ and male fitness $\bar{W}[r_O]$ can be expressed as a decreasing function of $r_O$, the unique Fisher-consistent value of $r_O$ is obtained by solving the equation $W[r_O] = \bar{W}[r_O] \cdot r$. This yields

$$r_O = \frac{1}{2\tilde{\mu}_1^2 n s^2} \tilde{n} \left( 2B\mu_1\tilde{\mu}_1 s\tilde{s} + A + M\sqrt{n}\sqrt{\tilde{n}}(B\tilde{s} + s(\tilde{s} - B\tilde{s} - 1)) \right) \sqrt{4B\mu_1\tilde{\mu}_1 s\tilde{s} + A} \quad (23)$$

where $A = M^2 n\tilde{n}(s - B\tilde{s} + (B-1)s\tilde{s})^2$ and $B = k/\tilde{k} \cdot n/\tilde{n} \cdot r$.

The same method of deriving $r_O$ also works for the model with post-mating parental investment, using fitness functions equations (1) and (2), and yielding equation (23) with $B = k/\tilde{k} \cdot n/\tilde{n} \cdot s/\tilde{s} \cdot r$.

With pre-mating parental investment, the expected parental investment and the realised parental investment in any given brood are the same; hence $\frac{\mathrm{d}\tau}{\mathrm{d}T} = 1$. Evolutionary trajectories calculated based on equations (21) and (22) are virtually indistinguishable from those in Fig. 1. This confirms that the timing of parental investment relative to mating does not fundamentally change the properties of the null model.

**Data availability.** This article contains no data.

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

## Acknowledgements

We thank Hanna Kokko for generous discussions, and in particular for the idea to attempt a unification of refs. 9, 13 and 19. We thank Jaakko Toivonen, Jonathan Henshaw and Sara Calhim for useful comments on the manuscript, and the Academy of Finland (L.F.; grant 283486) and the Australian Research Council (M.D.J.) for funding.

## Author contributions

L.F. contributed through modelling and writing. M.D.J. contributed through discussion of ideas and writing.
