## [Peer Review File · Nature Communications]

Reviewers' comments:

Reviewer #1 (Remarks to the Author):

This study investigates the causes of sex role divergences, focusing on sex differences in parental care. Although a large number of theoretical work have explored this question, our understanding of the causes of such sex differences is far from complete. One of the most comprehensive model of the problem was published by Kokko & Jennions (2008), that has become a highly cited paper with more than 250 citation since its publication (Web of Science, February 2016), and formed the basis of several later modelling and empirical work. The current study challenges this model by identifying a conceptual error in its original formulation, and after correcting that error provides a set of revised explanations and predictions. Most importantly, it shows that (1) sex roles do not converge towards equal care when anisogamy is the only difference between the sexes, and that (2) small initial sex differences in care can easily be amplified by evolutionary feedback between care and competition for mates. This study has the potential to interest a wide range of researchers and can be an important contribution to the understanding of sex role evolution.

Although the manuscript is clearly written and the results are presented in an accessible way in most of its parts, I have a number of questions and comments regarding the assumptions of the study and the interpretation of the results:

Lines 108-114: Synergistic benefits of biparental care were introduced into the model to get a result with stable parental care by both sexes. Why did the authors use this particular assumption (i.e. synergistic benefits), not some other, for this purpose? How realistic this assumption is? Please provide some background information that can highlight the exact reason for this step in the model construction.

Lines 123-127: Two traits model: A brief non-technical description of how coevolution between mating competition and care was modelled would be useful here. This part of the model is only described in the Methods section, but in a rather technical way, from that it is difficult to see exactly how changes in mating competition affect care and vice versa.

Lines 173-175: "This result contradicts findings of Kokko & Jennions 13 (see their Fig. 5), who reported that the proportion of care provided by males decreased when competing became costlier." - In a sense K & J say the reverse, i.e. that male care increases when competing becomes a costly behaviour due to male biased ASR (i.e. due to the presence of more competitors). See the last third part of their Fig. 5 where both male care and ASR/OSR are increasing. Please explain the basis of your interpretation.

Lines 182-185, two traits model: Here the authors show the conditions (reduced paternity, sexual selection) that can lead to the evolution of female only care, by creating small initial sex differences in care that are subsequently amplified by the feedback mechanism. However, it is not obvious from the current discussion of this feedback process how male only care can evolve in this system. For example, what factors can cause the initial sex difference (somewhat more care by males) that can be amplified later by the coevolutionary

process? This would be important to show because the authors criticize other theoretical work on the basis of its inability to explain reversed sex roles (line 203).

Lines 236-238: " This occurs because the MSR (which in the absence of sex differences in mortality is equivalent to the ASR) is inversely proportional to male reproductive value". Why is this only true for MSR, but not for ASR (irrespectively whether or not sex differences in mortality exist)? Please explain this briefly, because intuitively the difference is not obvious (even from the description of reproductive value as presented in the Appendix).

Lines 246-247: "Again, this is not due to a causal effect of the ASR. The pattern is better explained by low mortality making caring cheaper for males, since it reduces their probability of dying while caring (eq. A6)." - How can causality (or the lack of it) inferred from the model, and, particularly, exactly which results can be used to determine the better explanation from the two offered in this section? For example, why it cannot be that both decreased mating opportunity (due to ASR bias) and low cost of care (due to low mortality) contribute to the evolution of male care? (See also the next comment on causality).

Lines 252-254: "We corroborate our claim that the ASR is not an independent causal factor affecting male care by showing that the ASR is uncorrelated to the proportion of male care if it changes solely due to variation in time-in mortality (Fig. 3b, Fig. 4b)." - I am not fully convinced that ASR is not causally linked to sex roles. First, the number of available partners per male is very likely to have an influence on his mating prospects, that in turn is also likely to influence the relative costs and benefits of caring versus competing for mates. In my view, ASR can have a causal effect on sex roles in this way, for example through influencing the males' perception of mate availability. It is strange that this role of mate availability does not appear in the interpretations of the model's results, but is usually replaced by suggestions for alternative mechanism (e.g. preceding point above). Second, not only ASR can affect sex roles, and its effects may depend on other factors. For example, the cost of high mortality during mating competition may cancel the benefits from high mate availability (caused by female-biased ASR), and this may result in no correlation between ASR and male care. Thus, I am not sure that it is correct to use the lack of correlation under some circumstances as a proof for lack of causal relationship between ASR and sex differences in care.

Lines 276-279: "This could be particularly relevant in sexually size dimorphic species, where a positive correlation between body size dimorphism and ASR arises through sex differences in juvenile mortality and/or maturation rates 25,26." - This correlation should be negative by the logic presented here, because juvenile mortality is usually higher (and maturation is slower) in the larger sex, which means, for example, that a higher degree of sexual size dimorphism towards males should be associated with a lower (i.e. female-biased) ASR.

Lines 279-281: " Second, an ASR bias might arise if one sex generally suffers higher mortality, including while caring, which would reduce its propensity to care (Fig. 3a)" - It is unclear why the same high mortality during mating competition doesn't reduce its propensity to compete in a similar way. Please explain briefly.

Lines 574-592: Appendix 1: revising Kokko and Jennions' model: This part describes what is the error in Kokko & Jennions (2008). If I understand correctly, the authors of the current study suggest that the original model incorrectly assumes an "opportunity cost" of caring, because in an equilibrium population the timing of returning to the mating pool doesn't affect the pay-off from mating. However, it is unclear to me whether opportunity cost is used only in this sense, or instead in a more general way (including reduced mating opportunity due to reduced length of time spent in the mating pool) in its original formulation in K & J (2008). If K & J (2008) formulates opportunity cost in the latter, wider sense, then it is unclear how the critique based on timing (lines 585-589) can be relevant. I may be wrong on this point, but because this is a crucial part of the study, it would be particularly important to show very clearly that your interpretation of the opportunity cost of K & J (2008) is correct.

Lines 648-650: "In contrast, Kokko and Jennions 13 defined the number of surviving offspring per brood a priori, without accounting for demographic stability." - It is unclear why is the current approach better - why should be the population in equilibrium? Please explain this point.

Reviewer #2 (Remarks to the Author):

Review of Nature Communications NCOMMS-16-00777

I will begin by saying that I quite enjoyed reading this manuscript. This is a topic on which I am not overly familiar and I think the authors did an excellent job summarizing the current theoretical literature. I also feel that they have developed a thorough and general model that will make a nice contribution to this topic.

In this manuscript, the authors build on past theoretical work to investigate causes of sex-specific differences in levels of parental care. Specifically, they challenge conclusions from other recent theoretical work that claimed that there exists a negative feedback that will ultimately lead to equal investment in parental care between sexes. This hinges critically on the use of a synergistic model, where biparental care provides a benefit above and beyond the effects that would be expected if effort from both parents were additive. I do not have any major concerns with the work presented here. I found the general arguments convincing and intuitive. I only have a few small suggestions that I think might improve clarity and help ease readers through this fairly complex model.

Cheers,
-Leithen

Minor suggestions:

-Throughout I would avoid referring to parameter names directly in the text. For example, on L169, I would replace "because both μ_I and μ_O change" with "because both time-

in mortality rate, μ_I , and time-out mortality rate, μ_O , change". This greatly reduces the amount of jumping around one has to do while reading the paper and would, I believe, help a lot here.

-I think a table of parameter definitions would be helpful

-I would provide a discussion of the "Fisher Condition" in the Introduction.

L53: missing closing paranthesis

L174: replace "than" with "that"

L226: the writing here makes it sounds as though the "synergistic model" was constructed and used purely because the authors "needed a specific null model". I would suggest re-phrasing with a biological justification, or else empiricists might complain, as the authors say, about "the apparent ease with which modellers can overturn familiar predictions by altering a few key assumptions"

L400: replace "interpretation our" with "interpretation of our"

Fig. 4 is missing axes labels

Reviewer #3 (Remarks to the Author):

Comments to the authors

Fromhage and Jennions present a simple but insightful model of parental care evolution. They argue that in a demographically stable population there is no selection for biparental care even if the males and females are identical. They show that even a small deviance from male-female equality (e.g. in strength of sexual selection or in uncertainty of parentage) can quickly lead to uniparental care. They are also questioning the role of adult sex ratio in determining the pattern of care. First, their argument is based on Fisher condition, namely the reproductive value of both sex should be equal after weighting by the primary (or maturation) sex ratio (which they consider to be equal to the primary sex ration). From this it follows, that each sex should have equal number of reproduction because the reproductive value of an reproduction event is the same for both sexes. Because of this the argument by Kokko and Jennions (2008) fails, that is there is no selection pressure to equalise the care between sexes.

I found this paper presenting interesting results and clarifying the evolution of parental care further. Nevertheless, the paper is very

concise making it difficult to follow the authors' argument. I would also like the authors to discuss the relation between the strength of sexual selection and the investment in sexually selected traits (their third model). At the present they assume that k (strength of sexual selection) and x (investment in sex. selected trait) is independent from each other. On the other hand, I have the feeling that these two are related. A further point it might be worth clarifying of how the results are depend on the lack of senescence in the models.

Specific points

l. 26-28: "has repeatedly given rise to" should be "generally correlates with" because you cannot assume causation here, at least at the moment.

l. 28: Similarly, ""why does anisogamy" -> "does anisogamy".

l. 174: "reported than" -> "reported that"?

l. 221-222: Refer to the appendix here.

l. 226-227: Why do you need this kind of null model? Why isn't it enough to have the basic model (i.e. that without synergistic benefit of care)?

l. 238-242: I am not sure about this argument. It is true that an average male has low reproductive value if the MSR is male biased. But is this also true for a male qualified to mating? Please clarify.

l. 444: Does this equation means that an individual's fate of being qualified or non-qualified for mating is decided at once at the beginning of their life and remains so for the rest of its life?

l. 595-596: I am not sure about this equivalence. An individual entering into time-in after breeding will breed later for sure as it is qualified to do so. An individual entering at maturation have only a chance of $1/k$ to breed...

l. 611-612: Please explain why the consequences of caring a bit more after T must be calculated from the beginning of the care period.

l. Again, why do you need to take into account the probability of surviving to T if you assume that the female cares at T, i.e. she certainly survived until T.

Figure 4: Please label the vertical axes of the panels. Also, according to line 258 $\mu_O < \mu_I$; please indicate this in the figure legend.

Reviewer #4 (Remarks to the Author):

This paper purports to overturn the claim by Kokko & Jennions (2008; KJ hereafter) that negative feedback between the operational sex ratio (OSR) and the opportunity cost of care selects for egalitarian sex roles. If correct, this could be an important result, since the KJ paper is cited a lot and appears quite influential in guiding intuitions and interpretations of empiricists that study the role of the adult sex ratio (ASR) and OSR in mating system evolution.

I have several problems with this paper. First, the model has been written down in an incomplete and confusing fashion. One has to jump back and forth between the Methods section, the appendices and the KS paper to make sense of the model. Moreover, the two-trait model is based on an in-press paper which I did not have access too. It would be much preferable if the model were developed fully from scratch from start to finish.

Second, although the authors "trust that our current model is now Fisher consistent" (unlike the KJ paper), I believe that the current model is actually still inconsistent. Let me explain why:

The current model takes an approach which is superior to the KJ-approach by deriving sex-specific fitness functions (formulas (1) and (2) for lifetime reproductive success), and use these functions to derive marginal costs and benefits, instead of postulating formulas for costs and benefits based on plausible-sounding verbal arguments, as KJ did. However, it seems to me that the fitness functions are inconsistent. They must obey the "Fisher condition" that

$$W = rW_{\sim}, \quad (1)$$

Where

$$W = 1/k \cdot p/(1-sp)bS, \text{ with } p = a/(a + \mu_I), \quad W_{\sim} \text{ similar} \quad (2)$$

Note that strictly speaking this is not correct, since the authors neglected to take into account that the last period of parental care may be cut short by death. Therefore the last S is not always equal to $S(n_T + n_{\sim T_{\sim}})$, but may be smaller for the last period of parental care in an individual's life. Moreover, it is not the case that the expected value of $S(n_T + n_{\sim T_{\sim}})$ is equal to $S(\text{expected value of } n_T + n_{\sim T_{\sim}})$. But let's leave these details aside.

The sex-specific mating rates must also obey a "Fisher condition":

$$a/a_{\sim} = n/n_{\sim} r_0, \quad (3)$$

$$r_0 = r(1/k)L_I_{\sim} / ((1/k_{\sim})L_I) \quad (4)$$

Here r_O is the OSR and L_I the lifetime expected time-in, the formulas for which I had to find in KJ:

$$L_I = 1 / (a(1 - \exp(-\mu_O T)) + \mu_I) \text{ (similar for males) (5)}$$

In addition, for the population to be stable, we must have

$$b = k(1 - ps)(r + 1) / (pS). \text{ (6)}$$

When I combine the above equations (1) and (3), with the results from (2), (4)-(6) plugged in, and solve for a and a_{\sim} , I get

$$a = a_{\sim} \mu_I z / (a_{\sim}(1 - s_{\sim}) - a_{\sim}z(1 - s) + \mu_I I_{\sim})$$

where $z = r k / k_{\sim} n / n_{\sim}$

This appears to be inconsistent with the assumption that $a = Mn r_O^{(1/2)}$ and $a = Mnr_O^{(-1/2)}$. Actually, this assumption was made by KJ (p948) without much justification except equation (3) above. Any other power than 1/2 would have worked just as well, so it's not clear why 1/2 was chosen by KJ and the current model.

A revised version of the paper would have to do a better job of showing that all assumptions are consistent with each other and perhaps modify the model to make this possible in the first place.

Response to Reviewers' comments

Reviewer #1:

This study investigates the causes of sex role divergences, focusing on sex differences in parental care. Although a large number of theoretical work have explored this question, our understanding of the causes of such sex differences is far from complete. One of the most comprehensive model of the problem was published by Kokko & Jennions (2008), that became a highly cited paper with more than 250 citations since its publication (Web of Science, February 2016), and formed the basis of several later modelling and empirical work. The current study challenges this model by identifying a conceptual error in its original formulation, and after correcting that error provides a set of revised explanations and predictions. Most importantly, it shows that (1) sex roles do not converge towards equal care when anisogamy is the only difference between the sexes, and that (2) small initial sex differences in care can easily be amplified by evolutionary feedback between care and competition for mates. This study has the potential to interest a wide range of researchers and can be an important contribution to the understanding of sex role evolution.

Although the manuscript is clearly written and the results are presented in an accessible way in most of its parts, I have a number of questions and comments regarding the assumptions of the study and the interpretation of the results:

Lines 108-114: Synergistic benefits of biparental care were introduced into the model to get a result with stable parental care by both sexes. Why did the authors use this particular assumption (i.e. synergistic benefits), not some other, for this purpose? How realistic this assumption is? Please provide some background information that can highlight the exact reason for this step in the model construction.

Response: We have revised this section as suggested. It now reads:

“This prediction from our initial null model hinges on the assumption that broods benefit to the same extent from a given increment of care by either sex. To relax this assumption, we also present a synergistic null model in which the benefits of care from both sexes are synergistic rather than additive. Biologically, this is often appropriate. For example, it is likely whenever the sexes provide somewhat different forms of care that complement each other (e.g. food versus protection). In contrast to our initial null model, this synergistic null model predicts that a specific division of care between the sexes will evolve. This outcome additionally allows us to explore the effect that various factors of interest (e.g. sex-specific mortality rates) have on the evolution of the proportion of care provided by males.” (lines 115-124)

In brief, synergistic benefits were the most parsimonious, but still biologically plausible, way to produce a null model with stable biparental care. This feature is desirable because it improves the model's usefulness as conceptual tool for studying the evolution of parental care from first principles.

Reviewer #1: Lines 123-127: Two traits model: A brief non-technical description of how

coevolution between mating competition and care was modelled would be useful here. This part of the model is only described in the Methods section, but in a rather technical way, from that it is difficult to see exactly how changes in mating competition affect care and vice versa.

Response: Following this suggestion, we now write:

“We model this as a feedback loop in which parental care affects the evolution of competitiveness, competitiveness has mortality costs, and mortality affects the evolution of parental care.” (lines 139-141)

We realise that this is still very concise, but it would take too much space to describe the methods (especially in simple language) in detail in the main text. We are constrained by the format of *Nature Communications* whereby the *Methods* are after the *Results*. We hope, however, that in a model-based paper, such as ours, most readers who want details will read the *Methods* (where we provide full details) before the *Results*. As it stands, we think this is still sufficient information for a reader who reads the main text to follow the main argument of the paper.

Reviewer #1: Lines 173-175: "*This result contradicts findings of Kokko & Jennions 13 (see their Fig. 5), who reported that the proportion of care provided by males decreased when competing became costlier.*" - In a sense K & J say the reverse, i.e. that male care increases when competing becomes a costly behaviour due to male biased ASR (i.e. due to the presence of more competitors). See the last third part of their Fig. 5 where both male care and ASR/OSR are increasing. Please explain the basis of your interpretation.

Response: This seems to be a misunderstanding. In the last third of KJ's Fig. 5, they describe a change whereby *caring* becomes costlier than *competing*, rather than the other way round (it reads: "caring becomes dangerous"). Note that relative costliness here refers not to the ASR, but to the difference in mortality risk associated with either caring or being in the mating pool and competing for mates. KJ08 report that the proportion of male care increases when caring becomes costlier (one of the 'paradoxical' results). By contrast, where *competing* becomes costlier in KJ08's Fig. 5 (in the middle part), the proportion of male care decreases. Thus, it is correct for use to write that KJ08 "reported that the proportion of care provided by males decreased when competing became costlier". Our results do disagree with those of KJ08.

Reviewer #1: Lines 182-185, two traits model: Here the authors show the conditions (reduced paternity, sexual selection) that can lead to the evolution of female only care, by creating small initial sex differences in care that are subsequently amplified by the feedback mechanism. However, it is not obvious from the current discussion of this feedback process how male only care can evolve in this system. For example, what factors can cause the initial sex difference (somewhat more care by males) that can be amplified later by the coevolutionary process? This would be important to show because the authors criticize other theoretical work on the basis of its inability to explain reversed sex roles (line 203).

Response: We have added some discussion of this topic. The text reads:

“If anisogamy tends to favour the evolution of female-biased care, how can we explain male-only care in the light of our model? One possibility is that biparental care, driven by synergy, acts as an intermediate stage, followed by environmental changes that affect any of the parameters we have shown to be relevant in this context (the maturation sex ratio, the strength of sexual selection, and certainty of parentage). Alternatively, an environmental change could directly reverse the relative time-out durations of the sexes, as documented, for example, in Mormon crickets in response to diet quality²⁸.” (lines 368-375)

Specifically, we note that any factors that result in a male-biased sex ratio at maturation will select for greater male care, as will factors that reduce the mortality rate of caring males. So too would greater female than male uncertainty of parentage: but it is hard to conceive of a plausible mechanism whereby this would arise (aside from egg dumping by females whose eggs have been fertilised by the focal male). The main feature we are criticising in earlier models is an inherent assumption that anisogamy results in a ‘positive feedback’ for greater non-gametic parental investment by females, and less by males. This means that sex role reversal is not explicable, except by invoking additional *post hoc* factors. In contrast, our model provides an account of the factors that drive feedback loops. We do not assume there is always feedback between pre- and post-mating parental investment – otherwise we would never see cases where there is male-only care.

Reviewer #1: Lines 236-238: " *This occurs because the MSR (which in the absence of sex differences in mortality is equivalent to the ASR) is inversely proportional to male reproductive value*". Why is this only true for MSR, but not for ASR (irrespective whether or not sex differences in mortality exist)? Please explain this briefly, because intuitively the difference is not obvious (even from the description of reproductive value as presented in the Appendix).

Response: We made two changes to address this comment. Firstly, we expanded the following statement:

“The dependence of reproductive values on the MSR follows from the Fisher condition (Appendix 1). By contrast, a similar relationship between reproductive values and the ASR does not generally hold; adult mortality can affect the ASR without changing the sexes’ reproductive values. Intuitively, this occurs because, regardless of adult mortality, an even MSR imposes the constraint that maturing males must on average have the same number of offspring as maturing females. Thus, if (say) males suffer high adult mortality, this decreases their lifespan but simultaneously increases their reproductive rate while alive, generating no overall (dis-)advantage as compared to females. This is analogous to the well-known logic of Fisherian sex ratio evolution²⁶, whereby adult mortality generates no selection on the primary sex ratio as it makes neither sex more valuable than the other.” (lines 276-286)

Consider the following example: in a stable population with even MSR, the expected

(=average) number of maturing offspring per male must be 2 (one son and one daughter) – otherwise the population size would not be stable. The same holds for females. It follows that neither sex has higher fitness, and in this model fitness is equivalent to reproductive value in time-in (since individuals in time-in are indistinguishable from newly matured individuals). And this holds regardless of any ASR bias due to adult mortality. Secondly, we added the following statement in Appendix 1:

“The Fisher condition also implies that the reproductive value of males (relative to females) is inversely proportional to the MSR. This is true for average males and females (namely, $\frac{\tilde{v}}{v} = \frac{1}{r}$) and also for those qualified to mate (namely, $\frac{\tilde{v}\tilde{k}}{vk} = \frac{1}{r}\frac{\tilde{k}}{k}$). Since the right hand side of these equations depends only on r , k and \tilde{k} , it follows that the reproductive value of males relative to females is independent of adult mortality, and also of any associated variation in the ASR.” (lines 678-683)

Reviewer #1: Lines 246-247: "*Again, this is not due to a causal effect of the ASR. The pattern is better explained by low mortality making caring cheaper for males, since it reduces their probability of dying while caring (eq. A6).*" - How can causality (or the lack of it) inferred from the model, and, particularly, exactly which results can be used to determine the better explanation from the two offered in this section? For example, why it cannot be that both decreased mating opportunity (due to ASR bias) and low cost of care (due to low mortality) contribute to the evolution of male care? (See also the next comment on causality).

Response: We infer causality by analogy to processes that occur in the real world. For example, if we reduce male mortality during time-out ($\mu_{O\sim}$), this can be treated as analogous to performing an experimental manipulation of male mortality (e.g. remove predators): if it increases the ASR, we infer that the ‘arrow of causality’ runs in the direction (lower $\mu_{O\sim}$) \rightarrow (higher ASR).

Using the information that the cost of male care is proportional to $\mu_{O\sim}$ (eq. A6), we infer the following causal chain to describe the relationship between $\mu_{O\sim}$ and the proportion of male care:

(lower $\mu_{O\sim}$) \rightarrow (lower cost of male care) \rightarrow (higher proportion of male care)

Note that the ASR is not involved in this chain, despite itself being affected by $\mu_{O\sim}$. However, decreasing $\mu_{I\sim}$ (time-in mortality) does NOT affect the proportion of male care, despite having a similar effect on the ASR. Thus, a change in the ASR *per se* does not seem to select for a change in male care.

However, in our manuscript we do not wish to enter a philosophical discussion about the nature of causality. Instead, we toned down our previous claims, and added the following statement to clarify our position:

“Unfortunately, it turns out that this claim was misleading: our updated model no longer predicts any effect of the ASR *per se* that is independent of the source of

ASR variation. In other words, without specifying the source of ASR variation, we cannot predict the relationship between the ASR and the proportion of male care. We therefore consider it more appropriate to explain causal relationships, and formulate predictions, at the level of the underlying factors affecting the ASR, rather than based on the ASR itself.” (lines 261-267)

Reviewer #1: Lines 252-254: "*We corroborate our claim that the ASR is not an independent causal factor affecting male care by showing that the ASR is uncorrelated to the proportion of male care if it changes solely due to variation in time-in mortality (Fig. 3b, Fig. 4b).*" - I am not fully convinced that ASR is not causally linked to sex roles. First, the number of available partners per male is very likely to have an influence on his mating prospects, that in turn is also likely to influence the relative costs and benefits of caring versus competing for mates. In my view, ASR can have a causal effect on sex roles in this way, for example through influencing the males' perception of mate availability. It is strange that this role of mate availability does not appear in the interpretations of the model's results, but is usually replaced by suggestions for alternative mechanism (e.g. preceding point above).

Response: We have removed this statement. Moreover, to further clarify the significance and limitations of our predictions, we have added the statement:

“Finally, it should be noted that our model does not consider facultative parental care decisions and therefore cannot be tested based on phenotypically plastic shifts in the level of male care in response to, say, mate availability.” (Lines 312-314)

Reviewer #1: Second, not only ASR can affect sex roles, and its effects may depend on other factors. For example, the cost of high mortality during mating competition may cancel the benefits from high mate availability (caused by female-biased ASR), and this may result in no correlation between ASR and male care. Thus, I am not sure that it is correct to use the lack of correlation under some circumstances as a proof for lack of causal relationship between ASR and sex differences in care.

Response: Please see above for how we dealt with the issue of causality.

Reviewer #1: Lines 276-279: "*This could be particularly relevant in sexually size dimorphic species, where a positive correlation between body size dimorphism and ASR arises through sex differences in juvenile mortality and/or maturation rates 25,26.*" - This correlation should be negative by the logic presented here, because juvenile mortality is usually higher (and maturation is slower) in the larger sex, which means, for example, that a higher degree of sexual size dimorphism towards males should be associated with a lower (i.e. female-biased) ASR.

Response: The referee is correct. Thanks for spotting our mistake. We have changed the sentence as suggested.

Reviewer #1: Lines 279-281: "*Second, an ASR bias might arise if one sex generally suffers higher mortality, including while caring, which would reduce its propensity to care (Fig. 3a)*" - It is unclear why the same high mortality during mating competition doesn't

reduce its propensity to compete [authors' note: we believe the referee means 'care'] in a similar way. Please explain briefly.

Response: The reason is that mortality while caring affects the cost of caring (eq. A6), whereas mortality during mating competition doesn't. As shown in Figures 3b and 4b, mortality during mating competition does not affect the sexes' relative propensity to care. Rather than expanding on this point here, we have done so in the previous paragraph, where we have added the statement:

“The lesson from this is that time-in mortality does not affect the proportion of male care (Fig. 3b, Fig. 4b) because it does not affect the reproductive value of males compared to females (see above); nor does it affect the cost of male care (eq. A6) relative to female care (eq. A3).” (Lines 306-309)

Reviewer #1: Lines 574-592: Appendix 1: revising Kokko and Jennions' model: This part describes what is the error in Kokko & Jennions (2008). If I understand correctly, the authors of the current study suggest that the original model incorrectly assumes an "opportunity cost" of caring, because in an equilibrium population the timing of returning to the mating pool doesn't affect the pay-off from mating. However, it is unclear to me whether opportunity cost is used only in this sense, or instead in a more general way (including reduced mating opportunity due to reduced length of time spent in the mating pool) in its original formulation in K & J (2008). If K & J (2008) formulates opportunity cost in the latter, wider sense, then it is unclear how the critique based on timing (lines 585-589) can be relevant. I may be wrong on this point, but because this is a crucial part of the study, it would be particularly important to show very clearly that your interpretation of the opportunity cost of K & J (2008) is correct.

Response: The problem with K & J (2008) lies in their mathematical expressions, regardless of the words used to describe them. When KJ08 subtract their eq. (1) from their eq. (2) on page 933, this operation implies that spending more time with the current brood has costs in terms of opportunities lost because the carer did not return to the mating pool sooner. Regardless of what we choose to call it, the existence of such costs does not follow from (and is unfortunately incompatible with) the assumptions of their model. This is because, in a population that is constant over time, the payoff from entering the mating pool must also be constant over time: it makes no difference for a focal individual's future (from the time of entering onwards) if it enters now or later, provided that it does eventually do so.

Please note that this is not the only problem with KJ's formulation. In lines 647-653, we mention another one, namely their implicit assumption that the sequence of events: [brief time-in, long time-out, reproduction] yields a higher 'pay-off from desertion' than the sequence [long time-in, brief time-out, reproduction]. We believe this unfortunate situation arose because KJ relied overly on (in the words of Referee #4) "postulating formulas for cost and benefits based on plausible-sounding verbal arguments". To avoid this pitfall, we have derived our results with two independent methods, one of which does not require a priori identification of the cost and benefits of caring.

Reviewer #1: Lines 648-650: "*In contrast, Kokko and Jennions 08 defined the number of surviving offspring per brood a priori, without accounting for demographic stability.*" - It is unclear why is the current approach better - why should be the population in equilibrium? Please explain this point.

Response: The current approach is better because it is free of contradictions (i.e. it is internally consistent). Although KJ08 did not account for demographic stability when defining brood size, they implicitly assumed demographic stability in other parts of their model (in particular, the derivation of the OSR). We now mention this issue and write:

"This was inconsistent with their implicit assumption of demographic stability when deriving the OSR." (Lines 730-732).

Moreover, the alternatives to demographic stability (in a deterministic model) are exponential growth and decay, neither of which is biologically plausible over longer time spans. In short, it is good modelling practice to assume that population size is stable because this is closer to reality than the alternatives. Once this assumption is made, it is then necessary to ensure that the model is set up so that it actually happens – which means that one cannot arbitrarily fix all birth-death parameters.

Reviewer #2:

I will begin by saying that I quite enjoyed reading this manuscript. This is a topic on which I am not overly familiar and I think the authors did an excellent job summarizing the current theoretical literature. I also feel that they have developed a thorough and general model that will make a nice contribution to this topic.

In this manuscript, the authors build on past theoretical work to investigate causes of sex-specific differences in levels of parental care. Specifically, they challenge conclusions from other recent theoretical work that claimed that there exists a negative feedback that will ultimately lead to equal investment in parental care between sexes. This hinges critically on the use of a synergistic model, where biparental care provides a benefit above and beyond the effects that would be expected if effort from both parents were additive. I do not have any major concerns with the work presented here. I found the general arguments convincing and intuitive. I only have a few small suggestions that I think might improve clarity and help ease readers through this fairly complex model.

Throughout I would avoid referring to parameter names directly in the text. For example, on L169, I would replace "because both μ_I and μ_O change" with "because both time-in mortality rate, μ_I , and time-out mortality rate, μ_O , change". This greatly reduces the amount of jumping around one has to do while reading the paper and would, I believe, help a lot here.

Response: Changed as suggested.

Reviewer #2: I think a table of parameter definitions would be helpful

Response: Following this suggestion, we have included Table 1.

Reviewer #2: I would provide a discussion of the "Fisher Condition" in the Introduction.

Response: Following this suggestion, we have added the statement:

“The so-called “Fisher-condition”³⁰ is a consistency requirement that matings and offspring must be accounted for from the perspective of both sexes: because each diploid offspring in a sexual species has a mother and a father; and each heterosexual mating involves a male and a female. We clarify how the Fisher condition applies to the different life history stages at which the sex ratio is measured, namely birth, maturation, adults and in the ‘mating pool’ (OSR).” (Lines 77-82).

However, we have retained the detailed discussion of this topic in its original location (the Discussion), as it sheds light on our results that is especially important in explaining why our model produces different results from those of the earlier highly cited model of Kokko & Jennions 2008.

Reviewer #2: L53: missing closing parenthesis

Reviewer #2: L174: replace "than" with "that"

Response: Both corrected.

Reviewer #2: L226: the writing here makes it sounds as though the "synergistic model" was constructed and used purely because the authors "needed a specific null model". I would suggest re-phrasing with a biological justification, or else empiricists might complain, as the authors say, about "the apparent ease with which modellers can overturn familiar predictions by altering a few key assumptions"

Response: Changed as suggested. It now reads:

“... we also present a synergistic null model in which the benefits of care from both sexes are synergistic rather than additive. Biologically, this is often appropriate. For example, it is likely whenever the sexes provide somewhat different forms of care that complement each other (e.g. food versus protection). In contrast to our initial null model, this synergistic null model predicts a specific division of care between the sexes will evolve. This additionally allows us to explore the relationship between various factors and the proportion of care provided by males.” (Lines 117-124)

Reviewer #2: L400: replace "interpretation our" with "interpretation of our"

Response: Corrected.

Reviewer #2: Fig. 4 is missing axes labels

Response: Thank you. We have added the label “time-in mortality” to the x axis.

Reviewer #3:

Fromhage and Jennions present a simple but insightful model of parental care evolution. They argue that in a demographically stable population there is no selection for biparental care even if the males and females are identical. They show that even a small deviance from male-female equality (e.g. in strength of sexual selection or in uncertainty of parentage) can quickly lead to uniparental care. They are also questioning the role of adult sex ratio in determining the pattern of care. First, their argument is based on Fisher condition, namely the reproductive value of both sex should be equal after weighting by the primary (or maturation) sex ratio (which they consider to be equal to the primary sex ratio). From this it follows, that each sex should have equal number of reproduction because the reproductive value of an reproduction event is the same for both sexes. Because of this the argument by Kokko and Jennions (2008) fails, that is there is no selection pressure to equalise the care between sexes.

I found this paper presenting interesting results and clarifying the evolution of parental care further. Nevertheless, the paper is very concise making it difficult to follow the authors' argument. I would also like the authors to discuss the relation between the strength of sexual selection and the investment in sexually selected traits (their third model). At the present they assume that k (strength of sexual selection) and x (investment in sex. selected trait) is independent from each other. On the other hand, I have the feeling that these two are related. A further point it might be worth clarifying of how the results are depend on the lack of senescence in the models.

Response: We agree that the strength of sexual selection, as measured by our parameters k and k_{\sim} , is in reality likely to coevolve with other components of the system. Our reason for not modelling this explicitly is that there is a vast range of possibilities of how, exactly, this coevolution could work. (Some possibilities were explored by Lehtonen & Kokko 2011: Phil Trans R Soc B 367, 211–221.) Adding an additional x - k feedback would greatly complicate the model and, we believe, detract from its main focus. (It is, however, something we would like to look at in the future). One of our main goals is to keep our model comparable to its predecessor by Kokko and Jennions 2008. This then allows the reader to clearly see the major differences in outcomes and why they arise. Given the extent to which KJ08 has been cited, we think it is important to make the contrast in findings clear.

Importantly, the current model still reveals the qualitative effect of k and k_{\sim} on the evolution of care. This allows us to estimate how co-evolutionary changes in k and k_{\sim} would affect the system. For example, if a given evolutionary step towards female- care in the first panel of Fig 5 causes k_{\sim} to increase from 1.0 to 1.2, we can see in the second panel where selection would lead us onwards from there. We now directly mention the potential for x - k coevolution in the text, and we also make the simple point that the existing results allow us to see how changes in k (driven perhaps by changes x) affect levels of care by each sex. We now write:

“Although we did not formally model the coevolution of investment into competitive traits (x and \tilde{x}) and changes in the strength of sexual selection (i.e. k and \tilde{k}), it is likely that they will coevolve. It is, however, difficult to know their exact relationship.

The same holds for the relationship between sexual selection and the OSR²² and this could be a profitable line of future research. Nonetheless, our current model still allows us to determine how changes in sexual selection (arising for whatever reason, including changes in competitiveness) affect patterns of care. For example, the effect of stronger sexual selection on male care can be seen by comparing the left and right graphs in Figs 1, 2 and 5. (Lines 206-215)

As for the lack of senescence in the models, we find it difficult to make any general statement, since it could affect multiple model components. For example, as senescence reduces the reproductive value of an aging female, eq. A3 suggests she should then care more. On the other hand, if senescence increases her mortality rate while caring, eq. A3 suggests she should care less. Thus, the overall effect is likely to depend qualitatively on the exact mechanism by which senescence manifests itself. Again, this is an additional topic that is beyond the scope of the current paper, but would be a useful future extension of the model. At present we have not added anything about senescence to the text, but we would be happy to do so if requested.

Reviewer #3: l. 26-28: "has repeatedly given rise to" should be "generally correlates with" because you cannot assume causation here, at least at the moment.

Reviewer #3: l. 28: Similarly, ""why does anisogamy" -> "does anisogamy".

Response: Both changed as suggested.

Reviewer #3: l. 174: "reported than" -> "reported that"?

Response: Corrected.

Reviewer #3: l. 221-222: Refer to the appendix here.

Response: We originally referred to Appendix 1 in line 219 (now line 246), where it seemed more appropriate. Additionally, however, we have now included the phrase "see *Methods*" on line 249.

Reviewer #3: l. 226-227: Why do you need this kind of null model? Why isn't it enough to have the basic model (i.e. that without synergistic benefit of care)?

Response: One practical reason is that we cannot test the effect of various variables of interest (i.e. sex-specific mortality rates or k or x) on the proportion of male care based on a null model that does not predict any specific proportion of male care (recall that there is a neutral line of equilibria; see Fig 1a). However, we now also provide a biological motivation for the synergistic null model. It reads:

"This prediction from our initial null model hinges on the assumption that broods benefit to the same extent from a given increment of care by either sex. To relax this assumption, we also present a synergistic null model in which the benefits of care from both sexes are synergistic rather than additive. Biologically, this is often appropriate. For example, it is likely whenever the sexes provide somewhat different

forms of care that complement each other (e.g. food versus protection). In contrast to our initial null model, this synergistic null model predicts that a specific division of care between the sexes will evolve. This outcome additionally allows us to explore the effect that various factors of interest (e.g. sex-specific mortality rates) have on the evolution of the proportion of care provided by males.” (lines 115-124)

Reviewer #3: l. 238-242: I am not sure about this argument. It is true that an average male has low reproductive value if the MSR is male biased. But is this also true for a male qualified to mating? Please clarify.

Response: This is indeed the case. To clarify, we have added the the following statement in Appendix 1:

“The Fisher condition also implies that the reproductive value of males (relative to females) is inversely proportional to the MSR. This is true for average males and females (namely, $\frac{\tilde{v}}{v} = \frac{1}{r}$) and also for those qualified to mate (namely, $\frac{\tilde{v}\tilde{k}}{vk} = \frac{1}{r k}$). Since the right hand side of these equations depends only on r , k and \tilde{k} , it follows that the reproductive value of males relative to females is independent of adult mortality, and also of any associated variation in the ASR.” (lines 678-683)

Since we have defined $1/\tilde{k}$ as the proportion of maturing males that are qualified to mate, increasing the MSR causes a proportional increase in the total number of males qualified to mate. Their success in siring the next generation is then shared among more males, so each of them obtains a smaller share.

Reviewer #3: l. 444: Does this equation mean that an individual's fate of being qualified or non-qualified for mating is decided at once at the beginning of their life and remains so for the rest of its life?

Response: Yes, this is how we modelled it. As we write:

“some individuals belong to a class comprising $1/k$ of females and $1/\tilde{k}$ of males (at maturation), who are ‘qualified to mate’ and are therefore the only individuals that will have the opportunity to care.” (lines 453-455)

Reviewer #3: l. 595-596: I am not sure about this equivalence. An individual entering into time-in after breeding will breed later for sure as it is qualified to do so. An individual entering at maturation have only a chance of $1/k$ to breed...

Response: Here we are concerned with the decisions of a caring individual, which by definition must be qualified to mate. To clarify, we have reworded so that the statement can only refer to individuals that have bred (hence qualified to mate):

***“Thus, the payoff from re-entering the mating pool after breeding is constant over time”.* (line 658)**

Note, however, that it is not the case that an individual qualified to mate who enters

the mating pool will “breed later for sure”. This is because in our model there exists a risk of dying at any point.

Reviewer #3: l. 611-612: Please explain why the consequences of caring a bit more after T must be calculated from the beginning of the care period.

Response: To clarify, we added the statement:

“To see why this is necessary, consider the extreme case where T is so long that the carer is unlikely to survive until T . In this case, selection is weakened by the fact that any mutant with a slightly different T is unlikely to survive long enough for its mutant phenotype to become expressed.” (lines 690-693)

In other words, we have to weight selection due to any change in fitness associated with increasing T slightly by the likelihood that T is even reached. In practice, of course, T is rarely going to evolve to be so long that the likelihood of dying while caring is very high – nor does this weighting affect the location of equilibria. However, it is necessary to maintain the equivalence with the method based on derivatives of fitness functions (eqns. 3 and 4). Without this weighting, the same equilibria are reached by slightly different trajectories.

Reviewer #3: l. Again, why do you need to take into account the probability of surviving to T if you assume that the female cares at T , i.e. she certainly survived until T .

Response: See previous response.

Reviewer #3: Figure 4: Please label the vertical axes of the panels.

Response: In our version, the vertical axes were all labelled correctly, but the horizontal axes were only labelled with parameter symbols rather than words. Hence we have now added the label “time-in mortality” to the x-axes. If the referee really meant the vertical axes, then this problem might have been caused by a technical error during file conversion. We will keep an eye on this during the re-submission.

Reviewer #3: Also, according to line 258 $\mu_0 < \mu_1$; please indicate this in the figure legend.

Response: Thank you for noticing this. The statement the referee refers to was misleading. We have now clarified:

“i.e. $\tilde{\mu}_1 = \mu_1$ increases relative to $\mu_0 = \tilde{\mu}_0$ ” (lines 298)

Reviewer #4:

This paper purports to overturn the claim by Kokko & Jennions (2008; KJ hereafter) that negative feedback between the operational sex ratio (OSR) and the opportunity cost of care selects for egalitarian sex roles. If correct, this could be an important result, since the KJ paper is cited a lot and appears quite influential in guiding intuitions and interpretations of empiricists that study the role of the adult sex ratio (ASR) and OSR in mating system evolution. I have several problems with this paper. First, the model has been written down in an incomplete and confusing fashion. One has to jump back and forth between the Methods section, the appendices and the KJ paper to make sense of the model. Moreover, the two-trait model is based on an in-press paper which I did not have access too. It would be much preferable if the model were developed fully from scratch from start to finish.

Response: We would first note that the format of *Nature Communications*, where the Methods follow the Results, invariably leads to some confusion if a reader reads the paper in the strict sequence in which it is presented. The journal's target readership also means that the model presentation should reflect a balance between completeness, conciseness, and readability at different levels for different kinds of readers. We believe that delegating some information to appendices and references makes the flow of the main argument much easier to follow for those less interested in the technical details. Unfortunately, this does make the task harder for the current reviewer – sorry! We leave it to the Assistant Editor to decide if we should radically change our presentation, considering that reviewers 1, 2 and 3 felt the paper was well written. To reduce the reader's need to jump back and forth, however, we have now included a notational overview in table 1. We have also revised the manuscript such that now we usually refer to variables by name and by symbol, thus further reducing the reader's need to memorise the notation.

To address the concern of incompleteness, we have added Appendix 3, in which we now reproduce some (slightly revised) expressions from KJ08 for the reader's convenience.

Finally, the in-press paper mentioned by the referee has now appeared: doi: 10.1111/evo.12874 (*Evolution* 70: 617-624). While it uses the same method for modelling the evolution of competitive traits, it does not contain essential information for understanding the present model. We apologise for not making a copy available at the time of submission (when it was still in the pre-proof stage).

Reviewer #4: Second, although the authors "trust that our current model is now Fisher consistent" (unlike the KJ paper), I believe that the current model is actually still inconsistent. Let me explain why: The current model takes an approach which is superior to the KJ-approach by deriving sex-specific fitness functions (formulas (1) and (2) for lifetime reproductive success), and use these functions to derive marginal costs and benefits, instead of postulating formulas for costs and benefits based on plausible-sounding verbal arguments, as KJ did. However, it seems to me that the fitness functions are inconsistent. They must obey the "Fisher condition" that

$$W = rW_{\sim}, (1)$$

Where

$$W = 1/k p/(1-sp)bS, \text{ with } p = a/(a + \mu_{\sim}), W_{\sim} \text{ similar } (2)$$

Note that strictly speaking this is not correct, since the authors neglected to take into account that the last period of parental care may be cut short by death. Therefore, the last S is not always equal to $S(nT + n_{\sim}T_{\sim})$, but may be smaller for the last period of parental care in an individual's life.

Response: It is always awkward to tell a reviewer they are wrong, but in this case we have to do so. We did not neglect to take into account that the last care period may be cut short by death. We work with the expected care duration per brood (see lines 486-493), which accounts for the possibility that any given care period may be cut short by death. In case the referee doubts our brief derivation of this expectation, here we provide a slightly longer derivation that explicitly refers to the duration of interrupted care periods:

Females survive any given care period (once started) with probability $s = e^{-T\mu_o}$, in which case they provide care for duration T . They fail to survive with probability $1-s$, in which case they provide a partial care period of expected duration

$$T_{unfinished} = \frac{\int_0^T e^{-t\mu_o}(\mu_o t) dt}{\int_0^T e^{-t\mu_o}\mu_o dt}$$

Here, $e^{-t\mu_o}$ is the probability that the female is still alive at time t ; $\mu_o dt$ is the probability that she dies at t ; hence the whole expression represents a weighted sum of partial care periods (weighted by their probabilities), where the denominator is a normalising factor that ensures the probabilities sum to one. (They do not sum to one automatically, because not all care periods end in death.) Averaging across finished and unfinished care-periods, this yields the overall expectation

$$\tau[T] = sT + (1 - s)T_{unfinished} = \frac{1 - e^{-\mu_o T}}{\mu_o},$$

which is the same as the expression in lines 489-490.

Reviewer #4: Moreover, it is not the case that the expected value of $S(nT + n_{\sim}T_{\sim})$ is equal to $S(\text{expected value of } nT + n_{\sim}T_{\sim})$. But let's leave these details aside.

Response: This is true, but we have never claimed (nor implicitly assumed) this to be the case. Only $S(\text{expected value of } nT + n_{\sim}T_{\sim})$ is relevant in our model, since we have defined average offspring survival as a function of the average amount of care per brood. To justify this assumption, we have added the statement:

“This assumption ensures that offspring benefit equally from additional care received from either parent, which allows us to follow Kokko and Jennions¹³ in

focussing on other factors. (This is a deliberate simplification; in reality, there may be many reasons – including some related to sex-specific care durations – why offspring might benefit more from additional care by one sex or the other.)” (lines 481-486)

Reviewer #4: The sex-specific mating rates must also obey a "Fisher condition":

$$a/a_{\sim} = n/n_{\sim} r_{\sim} r_{\sim} O, \quad (3)$$

$$r_{\sim} O = r(1/k) L_{\sim} l_{\sim} / ((1/k_{\sim}) L_{\sim} l_{\sim}) \quad (4)$$

Here $r_{\sim} O$ is the OSR and $L_{\sim} l_{\sim}$ the lifetime expected time-in, the formulas for which I had to find in KJ:

$$L_{\sim} l_{\sim} = 1 / (a(1 - \exp(-\mu_{\sim} O T)) + \mu_{\sim} l_{\sim}) \quad (\text{similar for males}) \quad (5)$$

In addition, for the population to be stable, we must have

$$b = k(1 - ps)(r + 1) / (pS). \quad (6)$$

When I combine the above equations (1) and (3), with the results from (2), (4)-(6) plugged in, and solve for a and a_{\sim} , I get

$$a = a_{\sim} \mu_{\sim} l_{\sim} z / (a_{\sim}(1 - s_{\sim}) - a_{\sim} z(1 - s) + \mu_{\sim} l_{\sim})$$

where $z = r k / k_{\sim} n / n_{\sim}$

This appears to be inconsistent with the assumption that $a = Mn r_{\sim} O^{(1/2)}$ and $a_{\sim} = Mn r_{\sim} O^{(-1/2)}$. [...] A revised version of the paper would have to do a better job of showing that all assumptions are consistent with each other and perhaps modify the model to make this possible in the first place.

Response: We are happy to confirm the consistency of our model. The reviewer has made a simple human error. They have mixed up k and k_{\sim} in their equation (4). After correcting this, the referee’s final equation becomes consistent with their equation (3). Phew! We were worried for a moment. But we appreciate the opportunity this criticism provided for us to double check.

Another way to check the model’s consistency is by noting that the OSR can be calculated in two ways: either by eq. A16 (based on the Fisher condition as applied to mating rates) or by eq. A13 (based on the Fisher condition as applied to fitness). Reassuringly, both methods yield identical results (at least for the parameter values used in our paper, and presumably for all biologically meaningful parameter values). In other words: if we fit the OSR to make mating rates consistent, fitness becomes consistent automatically, and vice versa – as it should be. We have added new text to make this clearer. It reads:

“In Appendix 2, we also present an alternative method of deriving r_0 , based on applying the Fisher condition to fitness rather than mating rates. Both methods yield identical results, as should be the case in a fully consistent model.” (lines 466-469)

Reviewer #4: Actually, this assumption was made by KJ (p948) without much justification except equation (3) above. Any other power than 1/2 would have worked just as well, so it's not clear why 1/2 was chosen by KJ and the current model.

Response: Again, we respectfully disagree. If we substitute mating rates $a=Mn r_0^X$ and $a\sim Mn\sim r_0^{(-X)}$ into the referee's eq. (3), we obtain

$$r_0^{(2*X)} = r_0$$

Since r_0 is not necessarily zero or one, $X=1/2$ is the only power that makes this equation generally true.

REVIEWERS' COMMENTS:

Reviewer #1 (Remarks to the Author):

Thank you for the authors for responding to my comments, I found their corrections adequate and careful. I only have one final suggestion. In the revised manuscript, the authors toned down their statements on the effects of ASR on male care by (1) removing an earlier sentence that stated that "the ASR is not an independent causal factor affecting male care ..." (lines 252-254 in the original manuscript), and (2) also by changing one of their original sentences on causality to say that "our updated model no longer predicts any effect of the ASR per se that is independent of the source of ASR variation" (261-263 in the revised manuscript). I think these changes did a good job to express and interpret their findings. However, in the abstract they still left a sentence saying "We further argue that explanations invoking a causal link between the adult sex ratio and the proportion of care provided by males are misleading." I would suggest to modify this sentence in line with the above corrections, e.g. by saying "We further argue that our model does not predicts any effect of the ASR per se that is independent of the source of ASR variation", which would be more consistent with the conclusion presented in the main text.

Reviewer #3 (Remarks to the Author):

I think the authors did a good job, the paper is now much more understandable.

A specific point:

l. 205: "figs. a-d" -> "figs. 5a-d"

Reviewer #4 (Remarks to the Author):

I am satisfied that my main technical concerns were due to my own errors and mine alone, so I happily stand corrected and I apologize for making the authors go the extra mile in explaining it.

My overall feeling is now that the current version of the paper is quite acceptable for publication in this journal.

Response to referees' comments

Reviewer #1 (Remarks to the Author): Thank you for the authors for responding to my comments, I found their corrections adequate and careful. I only have one final suggestion. In the revised manuscript, the authors toned down their statements on the effects of ASR on male care by (1) removing an earlier sentence that stated that "the ASR is not an independent causal factor affecting male care ..." (lines 252-254 in the original manuscript), and (2) also by changing one of their original sentences on causality to say that "our updated model no longer predicts any effect of the ASR per se that is independent of the source of ASR variation" (261-263 in the revised manuscript). I think these changes did a good job to express and interpret their findings. However, in the abstract they still left a sentence saying "We further argue that explanations invoking a causal link between the adult sex ratio and the proportion of care provided by males are misleading." I would suggest to modify this sentence in line with the above corrections, e.g. by saying "We further argue that our model does not predict any effect of the ASR per se that is independent of the source of ASR variation", which would be more consistent with the conclusion presented in the main text.

Response: Changed as suggested. The sentence in question now reads: "We further argue that our model does not predict any effect of the adult sex ratio (ASR) that is independent of the source of ASR variation."

Reviewer #3 (Remarks to the Author): I think the authors did a good job, the paper is now much more understandable. A specific point: l. 205: "figs. a-d" -> "figs. 5a-d"

Response: Changed as suggested.

Reviewer #4 (Remarks to the Author): I am satisfied that my main technical concerns were due to my own errors and mine alone, so I happily stand corrected and I apologize for making the authors go the extra mile in explaining it. My overall feeling is now that the current version of the paper is quite acceptable for publication in this journal.

Response: Thanks, we appreciate your support.